# AMAG: Additive, Multiplicative and Adaptive Graph Neural Network For Forecasting Neural Activity

**Jingyuan Li**[1], **Leo Scholl**[1], **Trung Le**[1], **Pavithra Rajeswaran**[2], **Amy Orsborn**[1,2], **Eli Shlizerman**[1,3]
1. Department of Electrical & Computer Engineering,
2. Department of Bioengineering,
3. Department of Applied Mathematics
University of Washington, Seattle, WA
{jingyli6,lscholl,tle45,pavir,aorsborn,shlizee}@uw.edu

## Abstract

Latent Variable Models (LVMs) propose to model the dynamics of neural populations by capturing low-dimensional structures that represent features involved in neural activity. Recent LVMs are based on deep learning methodology where a deep neural network is trained to reconstruct the same neural activity given as input and as a result to build the latent representation. Without taking past or future activity into account such a task is non-causal. In contrast, the task of forecasting neural activity based on given input extends the reconstruction task. LVMs that are trained on such a task could potentially capture temporal causality constraints within its latent representation. Forecasting has received less attention than reconstruction due to recording challenges such as limited neural measurements and trials. In this work, we address modeling neural population dynamics via the forecasting task and improve forecasting performance by including a prior, which consists of pairwise neural unit interaction as a multivariate dynamic system. Our proposed model—Additive, Multiplicative, and Adaptive Graph Neural Network (AMAG)—leverages additive and multiplicative message-passing operations analogous to the interactions in neuronal systems and adaptively learns the interaction among neural units to forecast their future activity. We demonstrate the advantage of AMAG compared to non-GNN based methods on synthetic data and multiple modalities of neural recordings (field potentials from penetrating electrodes or surface-level micro-electrocorticography) from four rhesus macaques. Our results show the ability of AMAG to recover ground truth spatial interactions and yield estimation for future dynamics of the neural population.

## 1 Introduction

The ability to encode, decode, and interpret neural activity could facilitate novel applications targeted toward neural activity, such as Brain-Computer Interfaces (BCIs) [15, 50, 52, 28, 49]. A promising methodology for neural activity encoding and decoding is the use of artificial Deep Neural Networks (DNN) [53, 84, 78]. These approaches encode neural activity into latent neural representations, facilitating decoding and interpretation of neural signals and correlating the activity with other modalities, e.g., brain-to-text translation [72].

Most existing DNN architectures, such as Latent Factor Analysis via Dynamical Systems (LFADS) and its variants, infer latent dynamics from single-trial population activities by encoding the neural time series into a summarized representation from which the original time series could be regenerated, a procedure known as reconstruction task [53, 65, 51]. While effective in capturing low-dimensional latent neural manifolds, the task does not consider causal aspects of neural dynamics. In contrast,

37th Conference on Neural Information Processing Systems (NeurIPS 2023).

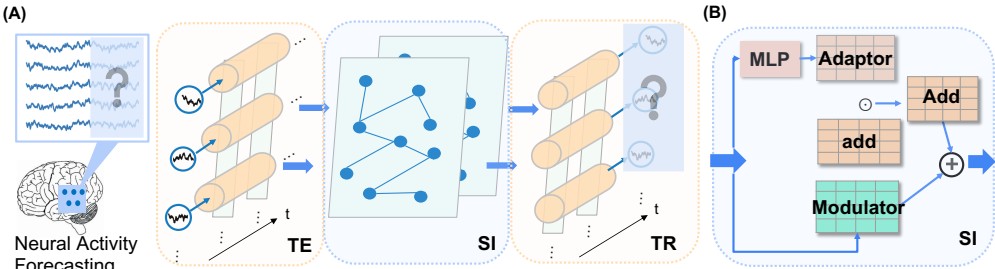

Figure 1: AMAG Overview. (A) The forecasting task and AMAG architecture with temporal Encoding (TE), Spatial Interaction (SI), and Temporal Readout (TR) modules to address the task. (B) Components of SI module.

the forecasting task, which aims to predict future activity based on past data, adhering to temporal causality constraints, could provide further insight into neural representations that are closer to representations employed by the brain [82, 55, 56, 18]. In addition, learning to forecast offers practical advantages since the signals that are predicted can help to reduce latency in real-time neural decoding applications (e.g., BCIs) or serve as a reference to detect real-time deviations of current recordings compared to earlier ones, indicating dynamic changes in the neuronal system or equipment malfunction [3].

Neural activity forecasting is inherently more challenging than reconstruction due to the requirement of a larger number of recorded neural signals, which are typically limited, and sensitivity to noise present in the signals. Incorporating priors into model structure design can be beneficial to ensure that instead of learning noisy information or overfitting the training data, the model learns features for forecasting future activity that are generalizable. A well-known prior is the spatial interaction among neurons, involving both addition [27, 9] and multiplication processes [19, 25] which could be modeled with Hodgkin-Huxley or Leaky Integrate-and-Fire [27, 1]. Previous works have utilized probabilistic graphical models to learn interactions [14, 42, 69, 36, 10] with a focus on capturing the neuron interaction itself rather than addressing downstream forecasting tasks. Models that learn the downstream task by considering spatial interaction have been proposed, such as Spatial Temporal Neural Data Transformer (STNDT) and Embedded Interaction Transformer (EIT), where the interaction between neurons is modeled using an attention mechanism [37, 44, 68]. Graph Neural Networks Networks (GNNs), designed to handle graph-structured data, are an alternative to the attention mechanism to model neuron interactions [75].

In this study, we propose a GNN with additive and multiplicative message-passing operations, called AMAG, for neuron signal forecasting. As shown in Fig. 1 (A), AMAG includes three modules: temporal encoding (TE), spatial interaction (SI), and temporal readout (TR). TE and TR model temporal dynamics for each recording channel independently. SI models the interaction between recording channels by leveraging the additive and multiplicative message-passing operations. We test the model on synthetic datasets and show the ability of AMAG to recover the ground truth interaction between channels. In addition, we demonstrate the effectiveness of the proposed model by testing it on two neural local field potential recording types (each with datasets from two animals). The first type is a publicly available dataset recorded by Utah array (penetrating probes) and the second type is a novel dataset of micro-electrocorticography ($\mu$ECoG) array recordings from the cortical surface. These four recordings are from the motor cortices of rhesus macaque monkeys performing reaching. We compare our model with other approaches, including GNN or non-GNN based. These computational evaluations demonstrate that AMAG outperforms other models regarding forecasting accuracy on neural recordings. We summarize the main contributions of our work below:

- *Modeling spatial-temporal relationship*. We present a graph neural network (GNN), called AMAG, that can forecast neuron activity by using sample-dependent additive and multiplicative message-passing operations with a learnable adjacency matrix.

- *Learning underlying spatial interaction*. We demonstrate the ability of AMAG to learn the underlying spatial interactions on synthetic datasets and show the importance of learning to forecast in discovering such interactions.

- *Generating future neural signals with AMAG.* We apply AMAG to four neural recording datasets and demonstrate its reliability in generating future neural activity while recovering the channel spatial proximity in the learned adjacency matrix and aligning neural trajectories in the latent space.

## 2    Related work

**Modeling Dynamics of Neural Signals.** Studying the dynamics of neural signals provides insights into the computation performed in the brain [70]. Neural dynamics has been modeled by Gaussian Process [73, 79, 81] and linear dynamical systems [45, 46, 23, 41, 2]. More recently, a series of nonlinear models have been proposed, including Multi-Layer Perceptron (MLP) [43, 83, 7], RNNs [52, 34, 57, 32, 60], Neural ODEs [35], and Transformers [77, 37, 44, 47]. Latent representations of population dynamics are often extracted while these models optimize for a reconstruction objective, upon which interpretable features emerge naturally [54, 58, 4]. A recent model that could forecast future neural activities is Preferential Subspace IDentification algorithm (PSID)[56, 55]. PSID proposes to forecast future neural activity along with behavior variables in two stages. In the first stage, the model learns to predict behavior and the behaviorally-relevant portion of neural signals. In the second stage, PSID learns to forecast both behaviorally-relevant and behaviorally-irrelevant neural signals [56, 55].

**Time Series Forecasting.** Time series forecasting, i.e., the prediction of next values in a time series, is a well-studied topic that encompasses various applications, such as traffic flow forecasting, motion prediction, and electric load forecasting. Examples of proposed methods include methods that are CNN based [31, 8], RNN based [24, 13, 26], Multi-layer perceptron [80] and so on. Transformers with additional components for capturing long-term dependencies have also been proposed for time series forecasting [68, 16, 74]. Also, models such as Graph Neural Networks (GNNs) have been proposed for time series forecasting focusing on datasets with inherent structural organization, e.g., traffic flow [30, 77, 38]. For example, these methods have been applied to functional Magnetic Resonance Imaging (fMRI) forecasting [71] leveraging Diffusion Convolutional Recurrent Neural Network (DCRNN) and Graph WaveNet (GWNet) [76, 40]. In these applications, GNN has been embedded in GRU gating mechanism to learn the spatial-temporal relationship (DCRNN), or CNN and GNN have been proposed to be intertwined to perform spatial-temporal modeling (GWNet). In contrast, GraphS4mer, which combines graph neural networks and structured state space models, performs temporal embedding followed by spatial message-passing in a sequential manner [66]. These methods perform only the additive message-passing, which may not be sufficient for a complex system such as the brain.

**GNNs for Neural Signals.** In recent years, GNNs have been proposed as plausible models for analyzing brain activity across different modalities [62, 33, 12, 29]. In non-invasive neural recordings, GNNs have been applied to EEG and fMRI data to capture the relationships between different brain regions or features. For example, LGGNet learns local-global-graph representations of EEG for various cognitive classification tasks [17], and BrainGNN has been proposed to identify neurological biomarkers with GNN, leveraging the topological and functional information of fMRI [39]. Recently, the method has been extended to analyze ECoG recordings as a graph diffusion process [59]. A few applications of GNNs to invasive neural measurements have been introduced, although these are less ubiquitous than GNNs for other recording types. The Two-Stream GNN model, which has been designed to classify anesthetized states based on ECoG recording by building two fixed graphs, is a notable example of such a model [11]. The first graph in this model is the functional connectivity matrix computed from the phase lag index, and the second is the dual version of the first one. The Two-Stream GNN is not suitable for forecasting since it combines temporal signals into a unified vector as the model input.

In light of the related work, the proposed AMAG is a GNN based approach which explicitly models interactions between neural signals to learn the dynamic features and interactions that can generate future neural signals. This is different from earlier models of learning neural dynamics with the primary focus on reconstruction without explicit modeling neural signal interactions. AMAG incorporates novel additive and multiplicative message-passing operations, which distinguish it from methods focusing on time series forecasting to associate GNNs with neural dynamics.

# 3 Methods

Our objective is to develop a model that estimates future neural dynamics given past activities. We define the relationship between future and past as $\hat{\boldsymbol{X}}_{t+1:t+\tau} = f_\theta(\boldsymbol{X}_{0:t})$, where $\boldsymbol{X} \in \mathbb{R}^{T \times C \times D}$ represents neural signals recorded from $C$ electrode channels over a time window of size $T$, with each channel having $D$ dimensional features. $\boldsymbol{X}_{0:t}$ denotes neural signals from 0 to $t$, and $\hat{\boldsymbol{X}}_{t+1:t+\tau}$ corresponds to predicted future neural signals for time window $t+1$ to $t+\tau$.

To estimate $f_\theta(\cdot)$, we formulate an optimization problem minimizing the loss $\mathcal{L}$ between predicted future signals $\hat{\boldsymbol{X}}_{t+1:t+\tau}$ and ground truth future signals $\boldsymbol{X}_{t+1:t+\tau}$ as follows

$$\min_{f_\theta} \mathbb{E}_{\boldsymbol{X}} \left[ \mathcal{L}(f_\theta(\boldsymbol{X}_{0:t}), \boldsymbol{X}_{t+1:t+\tau}) \right]. \tag{1}$$

Specifically, $f_\theta(\cdot)$ is constructed from three sub-modules as shown in Fig. 1: Temporal Encoding (**TE**) Module, Spatial Interaction (**SI**) Module, and Temporal Readout (**TR**) Module. The TE module captures temporal features for each channel individually. Then SI facilitates information exchange between channels, followed by TR to generate future neural activities. We describe the three modules in detail in the sections below.

## 3.1 SI with *Add* and *Modulator*

SI is constructed as a GNN. We represent a Graph $\mathcal{G} = (\mathcal{V}, \mathcal{E}, \boldsymbol{A})$, where $\mathcal{V}$ denotes a set of nodes $|\mathcal{V}| = C$, $\mathcal{E}$ is a set of edges, and weighted adjacency matrix $\boldsymbol{A} \in \mathbb{R}^{C \times C}$ contains the weights of edges. The neighborhood of a node $v$ is written as $\boldsymbol{N}(v)$, where $\boldsymbol{N}(v) = \{u | u \in V, (u, v) \in \mathcal{E}\}$. We denote neural signal from channel $v$ at time $t$ as $X_t^{(v)}$. $FC$, $MLP$ and $\sigma(\cdot)$ in the following sections represent a one-layer neural network, multi-layer perceptron, and sigmoid function, respectively.

Considering two types of interaction between neurons, additive [27] and multiplicative interactions [19, 25], we design the SI module to incorporate two sub-graphs for additive and multiplicative operations, namely Add and Modulator modules, respectively. In addition, we introduce a sample-dependent matrix, called Adaptor, to adjust the Add for each input sample, accounting for the complexity and dynamic nature of the brain as a system.

**Add Module**. The Add Module performs the additive interaction between channels with the message-passing function $\mathcal{M}_a(\cdot)$, depending on the adjacency matrix $\boldsymbol{A}_a \in \mathbb{R}^{C \times C}$. Assuming we have $d$ dimensional channel features $\boldsymbol{h}_t^{(v)}$ for node $v$ at timestep $t$ ($\boldsymbol{h}_t^{(v)} \in \mathbb{R}^d$), $\mathcal{M}_a(\cdot)$ updates $\boldsymbol{h}_t^{(v)}$ with the additive message from neighbor node $u$ as $\mathcal{M}_a(\boldsymbol{h}_t^{(u)}, \boldsymbol{h}_t^{(v)}, \boldsymbol{A}_a) = A_a^{(u,v)} \boldsymbol{h}_t^{(u)}$, where $u \in \boldsymbol{N}_a(v)$. Such that the updated channel feature of Add Module is the weighted feature of neighborhood,

$$\boldsymbol{a}_t^{(v)} = \sum_{u \in \boldsymbol{N}_a(v)} A_a^{(u,v)} \boldsymbol{h}_t^{(u)}. \tag{2}$$

The element $(u, v)$ in the adjacency matrix $\boldsymbol{A}_a$ indicates the influence of channel $u$ on $v$. Since $\boldsymbol{A}_a$ is shared for all the input sequences, while channel interaction can change across inputs, we introduce a sample-dependent *Adaptor* Module. We treat $\boldsymbol{A}_a$ as a fundamental adjacency matrix, and then we further learn a sample-dependent Adaptor as a matrix $\boldsymbol{S} \in \mathbb{R}^{C \times C}$ to uniquely modulate $\boldsymbol{A}_a$ depending on each sample. Then, the shared message-passing function Eq. 3 becomes a sample-dependent message-passing function,

$$\boldsymbol{a}_t^{(v)} = \sum_{u \in \boldsymbol{N}_a(v)} S^{(u,v)} A_a^{(u,v)} \boldsymbol{h}_t^{(u)}. \tag{3}$$

$S^{(u,v)}$ represents interaction strength between channel $u$ and $v$ relying on temporal embeddings for each channel, e.g., $\boldsymbol{H}^{(v)} = [\boldsymbol{h}_1^{(v)}, \ldots, \boldsymbol{h}_t^{(v)}]$ where $t$ is the context window for future neural signal generation. Specifically, we compute $S^{(u,v)}$ as $\sigma(MLP([\boldsymbol{H}^{(u)}, \boldsymbol{H}^{(v)}]))$, such that $S^{(u,v)}$ ranges from 0 to 1, similar to matrix construction for learning interpretable GNNs [48].

Since the fundamental adjacency matrix for the Add Module is unknown, $\boldsymbol{A}_a$ needs to be learned during the optimization of the model parameters. We observe that direct learning of $\boldsymbol{A}_a$ from random initialization could make the training process unstable and thus instead, we initialize the $\boldsymbol{A}_a$ with the precomputed correlation matrix from the neural recordings to stabilize the learning process.

**Modulator Module**. The Modulator Module is the multiplicative message-passing operator, with function $\mathcal{M}_m(\cdot)$. We incorporate the adjacency matrix $\boldsymbol{A}_m \in \mathbf{R}^{C \times C}$ to encode the neighborhood information that performs a multiplicative operation. $\mathcal{M}_m(\cdot)$ indicates how the feature of the neighborhood modulates the target node feature, which can be expressed as $\mathcal{M}_m(\boldsymbol{h}_t^{(u)}, \boldsymbol{h}_t^{(v)}, \boldsymbol{A}_m) = A_m^{(u,v)} \boldsymbol{h}_t^{(u)} \odot \boldsymbol{h}_t^{(v)}$, where $\odot$ represents Hadamard product. The Modulator Module includes all the multiplicative information from the neighbors to update feature for target channel $u$ at time $t$ as $\mathbf{m}_t^{(u)}$

$$\boldsymbol{m}_t^{(v)} = \sum_{u \in \boldsymbol{N}_m(v)} \mathcal{M}_m(\boldsymbol{h}_t^{(u)}, \boldsymbol{h}_t^{(v)}, \boldsymbol{A}_m) \tag{4}$$

Similarly to the adjacency matrix for the Add Module, the adjacency matrix in the Modulator Module is also trainable and is initialized from the correlation matrix.

In summary, at a specific timestep $t$, the output of SI is

$$\boldsymbol{z}_t^{(v)} = \beta_1 \boldsymbol{h}_t^{(v)} + \beta_2 FC(\boldsymbol{a}_t^{(v)}) + \beta_3 FC(\boldsymbol{m}_t^{(v)}), \tag{5}$$

with $\beta_1$, $\beta_2$, and $\beta_3$ controlling the contributions of the self-connection, Add Module, and Modulator Module respectively.

## 3.2 Temporal Processing with TE and TR Modules

TE and TR perform temporal encoding and decoding. In particular, TE embeds input neural signals $\boldsymbol{X}^{(v)} \in \mathbb{R}^{T \times D}$ from channel $v$ into the embedding space $\boldsymbol{H}^{(v)} \in \mathbb{R}^{T \times d}$. For multi-step forecasting, we mask neural signals after the context window $t$ with constant, i.e., $\boldsymbol{X}_{t:T}^{(v)} = constant$. Then SI updates representation of channel $v$ as $\boldsymbol{Z}^{(v)} \in \mathbb{R}^{T \times d}$, followed by TR generating future neuron activity $\hat{\boldsymbol{X}}_{t:T}^{(v)}$ taking $\boldsymbol{Z}^{(v)}$ as inputs. TE and TR can be Transformer or GRU. For further details on TE and TR, see Appendix A.2.

**Parameter Learning**. In summary, AMAG estimates future neural signals with $f_\theta$ implementing TE, SI, and TR sequentially to the past neural signals. The optimal parameters $\theta$ are obtained through backpropagation minimizing the Mean Squared Error (MSE) loss,

$$\mathcal{L} = \mathbb{E}_{\boldsymbol{X}} \left( \left\| \hat{\boldsymbol{X}}_{t:T} - \boldsymbol{X}_{t:T} \right\|_2 \right). \tag{6}$$

# 4 Experiments

## 4.1 Learning The Adjacency Matrix of a Synthetic Dataset

**Synthetic Dataset.** To explore the ability of AMAG to capture channel interactions, we generate synthetic datasets using linear non-Gaussian dynamic systems similar to previous works [61, 5, 6]. Specifically, with adjacency matrix $\boldsymbol{A}$ describing the spatial interactions among variables (channels), datasets $\boldsymbol{X}$ are generated recursively at each time step as $X_t = X_{t-1} + \mathcal{G}(X_{t-1}, \boldsymbol{A}) + \mu$, where $\mu$ is uniformly distributed noise term. We consider two scenarios where the connection ratio of the adjacency matrix is 0.2 and 0.5, respectively. The resulting datasets consist of multichannel time series, where the value of each channel at a given time step depends on its past and its neighboring channels, as defined by the adjacency matrix.

**Adjacency Matrix Recovery with AMAG.** We evaluate the ability of AMAG to recover the ground truth adjacency matrix and its dependency on tasks. We do so by additionally incorporating reconstruction and masked input reconstruction tasks as supplementary training objectives together with one-step and multi-step and forecasting tasks. The learned adjacency matrices, along with the ground truth matrix for 20 channels, are visualized in Fig.2 (Additional results with different numbers of channels in Appendix B.1). To assess the accuracy of recovering the underlying ground truth adjacency matrix, we employ the F1 Score as a quantitative measure. Our results show that the one-step forecasting task achieves the highest recovery of ground truth edges ($F1 = \mathbf{0.88}$ at 0.5 connection ratio), followed by the multi-step forecasting task ($F1 = \mathbf{0.82}$). This is compared to the reconstruction task which yields a lower F1 score of $\mathbf{0.63}$. The lower performance of reconstruction tasks could be due to the model's access to future signals, which potentially alleviates the need to learn the interactions with neighbors that future signals rely on. While multi-step forecasting generates

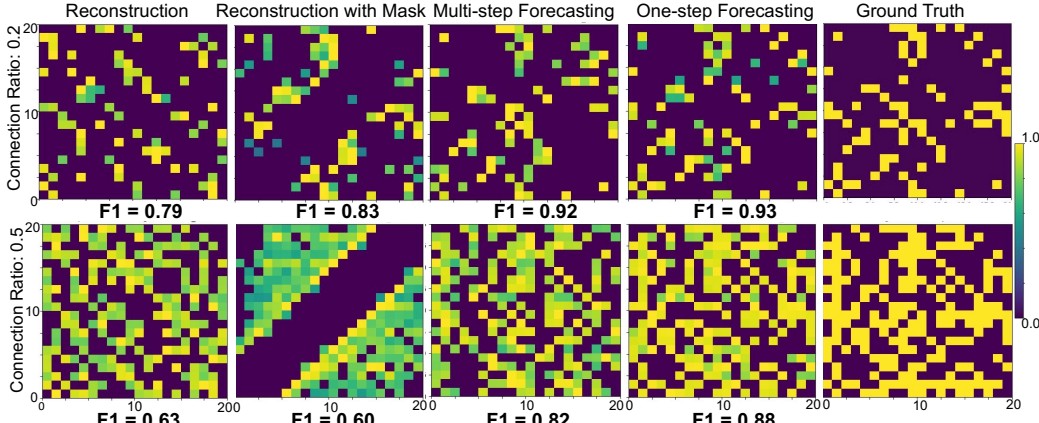

Figure 2: Colormap visualization of AMAG learned adjacency matrices arranged by F1 score (right better), with yellow indicating Ground Truth recovery while gree-blue indicating lower accuracy, as reconstruction task (first column), reconstruction of masked input task (second column), multi-step forecasting (third column) and one-step forecasting task (fourth column), alongside ground truth adjacency matrices generating synthetic data (last column), demonstrating AMAG's ability to recover the underlying spatial interactions doing forecasting tasks.

multiple future signals, some adjacency matrices may generate more accurate signals than others at a given moment, but considering all future moments collectively, different adjacency matrices may achieve similar performance, resulting in learned adjacency matrices being less accurate than one-step forecasting.

## 4.2 Evaluation on Forecasting with Neural Signals

**Dataset and Preprocessing** In this study, we collect neural signals from monkey Affogato (A), and Beignet (B) with $\mu$ECoG implanted in motor cortices, covering five different subregions (Primary Motor Cortex (M1), Premotor Cortex, Frontal Eye Field (FEF), Supplementary Motor Area (SMA), and Dorsolateral Prefrontal Cortex (DLPFC)) with 239 effective $\mu$ECoG electrodes arrays which have been used in [67]. During data collection, the subjects performed a reaching task towards one of eight different directions. All procedures were approved by the University of Washington Institutional Animal Care and Use Committee. In the following experiments, we include all electrodes from Monkey A and electrodes in M1 from Monkey B (More in the Appendix B.2). We additionally apply AMAG on two public datasets with field potential recorded with Utah arrays from the M1 of Monkey Mihili (M) and Chewie (C) where subjects performed similar hand-reaching task [22, 21]. M and C datasets contain the computed Local Motor Potential (LMP) and power band signal from eight frequency bands: 0.5–4 Hz, 4–8 Hz, 8–12 Hz, 12–25 Hz, 25–50 Hz, 50–100 Hz, 100–200 Hz, and 200–400 Hz. The resulting signal is downsampled to 30ms. Full broadband data from A and B were collected at 25 kHz. We then applied a preprocessing procedure to extract LMP and powerband signals from A and B datasets as used in M and C datasets [20, 64]. The powerband signals are included as additional feature dimensions. In experiments, we treat a target reaching trial as a sample for all four datasets. On M and C datasets, trials are temporally aligned to detect movement onset and include 300 ms before to 450 ms after movement onset. For the A and B datasets, trials are temporally aligned to the appearance of the peripheral target and include 600ms after the peripheral target onset.

**Benchmark Methods,** We compare AMAG with existing approaches, including variants of RNNs, Transformers, and GNNs. Specifically, we compare with the following methods: *LRNN*, RNN model without a non-linear activation function and gating mechanism [63]; *RNNf*, RNN model with non-linear activation and gating mechanism [32, 47]; *LFADS*, RNN based model with pre-computed initial states [65]. *NDT*, a decoder based transformer in temporal domain to estimate neural firing rate [78]; *STNDT*, a model that is similar to NDT with additional attention based spatial interaction between neurons [37]; *TERN*, a model that includes the GRU for encoding and temporal Transformer for decoding [32, 47]. *RNN PSID*, RNN based model with 2-stage training for learning behavior-related neural space and behavior-unrelated neural space combined to ensure accurate

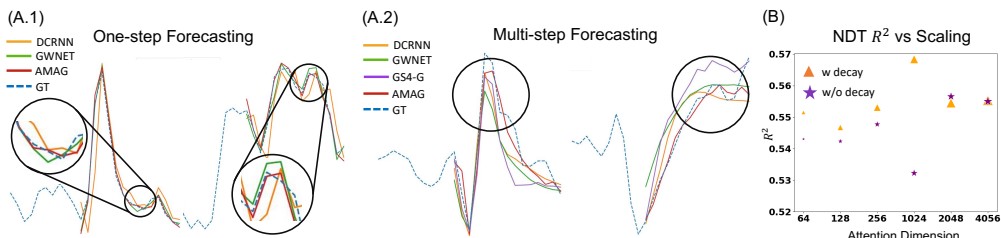

Figure 3: Panel A: Examples of predicted neuron trajectory for one-step (A.1) and multi-step (A.2) with representative methods DCRNN (orange), GWNet (green), GS4-G (purple), and AMAG (red). Ground Truth (GT) trajectories are shown in dashed blue. Panel B: Evaluation of the effect of attention dimension on NDT performance, with weight decay (w decay) in orange triangles, without weight decay (w/o decay) in purple circles. The larger size of the marker reflects the larger size of the model.

Table 1: *One-step forecasting results on Monkey M, C, B, and A with benchmark methods and AMAG*.

| | Monkey M | | | Monkey C | | | Monkey B | | | Monkey A | | |
|---|---|---|---|---|---|---|---|---|---|---|---|---|
| | $R^2\uparrow$ | $Corr\uparrow$ | $MSE\downarrow$ | $R^2\uparrow$ | $Corr\uparrow$ | $MSE\downarrow$ | $R^2\uparrow$ | $Corr\uparrow$ | $MSE\downarrow$ | $R^2\uparrow$ | $Corr\uparrow$ | $MSE\downarrow$ |
| STNDT [37] | 0.778±4e-4 | 0.883±4e-4 | 0.0333±1e-4 | 0.860±3e-3 | 0.928±1e-3 | 0.0111±2e-4 | 0.857±2e-3 | 0.932±8e-4 | 0.0089±2e-4 | 0.879±1e-3 | 0.939±7e-4 | 0.0074±7e-5 |
| NDT [78] | 0.837±2e-3 | 0.915±2e-3 | 0.0292±2e-3 | 0.908±4e-4 | 0.953±1e-4 | 0.0074±3e-5 | 0.897±3e-4 | 0.950±1e-4 | 0.0058±2e-5 | 0.924±6e-4 | 0.962±2e-4 | 0.0046±4e-5 |
| LFADS [65] | 0.847±3e-4 | 0.922±2e-4 | 0.0274±8e-5 | 0.905±7e-4 | 0.953±2e-4 | 0.0074±3e-5 | 0.846±1e-4 | 0.923±1e-4 | 0.0088±9e-5 | 0.903±7e-4 | 0.951±2e-4 | 0.0060±5e-5 |
| RNN [47] | 0.823±2e-4 | 0.911±1e-4 | 0.0250±4e-5 | 0.886±7e-4 | 0.943±3e-4 | 0.0091±5e-5 | 0.909±6e-4 | 0.954±2e-4 | 0.0052±4e-5 | 0.926±7e-4 | 0.963±2e-4 | 0.0045±4e-5 |
| RNN PSID [56] | 0.883±1e-4 | 0.940±1e-4 | 0.0108±2e-5 | 0.915±3e-4 | 0.957±6e-5 | 0.0071±2e-5 | 0.915±7e-4 | 0.957±3e-4 | 0.0048±5e-5 | 0.908±3e-4 | 0.953±4e-4 | 0.0049±1e-5 |
| LRNN [63] | 0.879±7e-4 | 0.937±4e-4 | 0.0137±7e-5 | 0.916±2e-4 | 0.957±7e-5 | 0.0067±1e-5 | 0.916±8e-4 | 0.957±4e-4 | 0.0047±4e-5 | 0.927±3e-4 | 0.963±8e-5 | 0.0045±2e-5 |
| TERN [47] | 0.866±7e-4 | 0.932±4e-4 | 0.0247±3e-4 | 0.920±9e-4 | 0.960±2e-4 | 0.0062±4e-5 | 0.888±1e-3 | 0.945±7e-4 | 0.0067±2e-4 | 0.929±4e-4 | 0.964±2e-4 | 0.0043±2e-5 |
| DCRNN [40] | 0.956±3e-3 | 0.978±1e-3 | 0.0190±3e-4 | 0.965±3e-3 | 0.983±2e-3 | 0.0026±3e-4 | 0.964±1e-4 | 0.982±8e-5 | 0.0020±4e-6 | 0.977±2e-3 | 0.988±8e-4 | 0.0014±1e-4 |
| GWNet [76] | **0.971±5e-4** | **0.986±4e-4** | **0.0176±6e-4** | **0.985±3e-4** | **0.992±1e-4** | **0.0012±2e-5** | 0.942±2e-3 | 0.971±1e-3 | 0.0033±1e-4 | 0.949±2e-3 | 0.974±1e-3 | 0.0031±1e-4 |
| **AMAG** | 0.965±1e-3 | 0.982±1e-3 | 0.0209±1e-3 | 0.972±1e-3 | 0.986±6e-4 | 0.0021±8e-5 | **0.973±2e-3** | **0.986±1e-3** | **0.0015±1e-4** | **0.979±7e-4** | **0.990±4e-4** | **0.0013±4e-5** |

future forecasting accuracy [56]; Notably, we do not compare with RNN PSID [56] for multi-step forecasting comparison since learning behavior-related neural space in the first stage of RNN PSID requires access to all the previous steps at a given moment, which is not available in a multi-step forecasting scenario. In addition, we compared AMAG with GNN based models. In particular, *GWNet*: A model which intertwines CNN and GNN for spatial interaction with a learnable adjacency matrix [76], *DCRNN*: embedded GNN in each GRU processing step using the predefined adjacency matrix [40], *GraphS4mer*: Sequential learning of temporal embedding and spatial interaction with time-dependent adjacency matrices [66]. For GraphS4mer we consider two variants, GRU based temporal learning (GS4-G) and state space model based temporal learning (GS4-S). Both variants are not included in one-step forecasting comparison, considering GraphS4mer's design of learning dynamic graphs for temporal windows, which increases computation complexity and is unnecessary in one-step forecasting.

**Experiment Setup.** For all four datasets, we generate future neural signals given at least 5 steps as the context corresponding to the 150ms time window. For one-step prediction, we use GRU architecture for TE and TR, and for multi-step prediction, Transformers are used for TE and TR (multi-step forecasting performance with GRU in Appendix B.4). All the models are optimized by Adam Optimizer on Titan X GPU. The initial learning rates are $1e-4$ and $5e-4$ for non-GNN methods and GNN based methods. We include three evaluation metrics. R-squared ($R^2$) measures the proportion of variance in the future neuron recordings that the forecasted signal can explain. Correlation ($Corr$) focuses on the matching of the trend of the ground truth signal and the forecasted signal. Mean squared error ($MSE$) measures the L2 distance of the forecasted signal to the ground truth neuron signal. Further details are described in Appendix B.3.

**Forecasting Performance.** We evaluate the performance AMAG to forecast future neural activities and present the results in Table 1 and Table 2, for one-step and multi-step forecasting, respectively. The standard deviation is obtained from three runs. Compared with the best non-GNN methods, AMAG improves the $R^2$ score by 8.2%, 5.0%, 5.7%, and 5.0% on Monkey M, C, B, and A datasets, respectively in Table 1. Transformer only methods, STNDT [37] and NDT [78] achieve less optimal performance on four datasets. For instance, on Monkey M, the $R^2$ score is 0.883 for the best RNN based methods (RNN PSID) and 0.837 for the best Transformer based method (NDT). This could be due to the fact that the neural signals are continuous and future signals are primarily dependent on nearby precedents, which fits the RNN use cases better. Consequently, the advantage of the Transformer encoding long-term dependency may not be as prominent in this context. While Graph methods achieve comparable performance as shown in Table 1, AMAG shows consistent performance

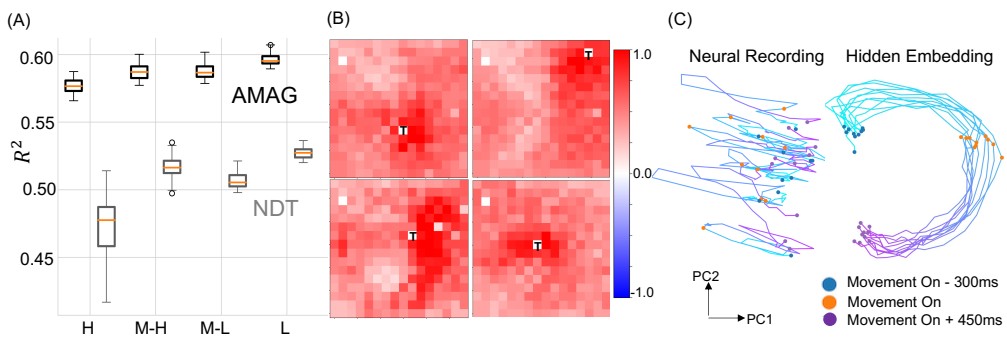

Figure 4: (A): Performance evaluation of masking channels from High (H), Mid-high (M-H), Mid-low (M-L), and Low (L) groups categorized by weight of learned adjacency matrix, showing the importance of high weight channel for forecasting performance. (B): Visualization of interaction strength of target channel to the reaming channels arranged according to spatial proximity in micro-ECoG array. (C): Examples of neural population trajectories in 2-dimensional PCA space projected from original neuron recording (left) and hidden embedding of AMAG (right).

Table 2: ***Multi-step forecasting*** *results on Monkey M, C, B, and A with benchmark methods and AMAG.*

| | Monkey M | | | Monkey C | | | Monkey B | | | Monkey A | | |
|---|---|---|---|---|---|---|---|---|---|---|---|---|
| | $R^2 \uparrow$ | $Corr \uparrow$ | $MSE \downarrow$ | $R^2 \uparrow$ | $Corr \uparrow$ | $MSE \downarrow$ | $R^2 \uparrow$ | $Corr \uparrow$ | $MSE \downarrow$ | $R^2 \uparrow$ | $Corr \uparrow$ | $MSE \downarrow$ |
| LFADS [65] | 0.233±4e-3 | 0.485±3e-3 | 0.0780±4e-4 | 0.514±3e-3 | 0.725±2e-3 | 0.0387±2e-4 | 0.427±4e-3 | 0.672±6e-3 | 0.0338±4e-4 | 0.731±2e-3 | 0.855±1e-3 | 0.0164±1e-4 |
| STNDT [37] | 0.227±3e-3 | 0.480±4e-3 | 0.0780±4e-4 | 0.518±1e-3 | 0.720±5e-4 | 0.0387±2e-4 | 0.525±4e-3 | 0.725±2e-3 | 0.0288±2e-4 | 0.726±9e-3 | 0.852±6e-3 | 0.0167±6e-4 |
| LRNN [63] | 0.250±3e-3 | 0.500±2e-3 | 0.0757±1e-4 | 0.483±4e-3 | 0.697±2e-3 | 0.0416±2e-4 | 0.507±6e-3 | 0.725±1e-3 | 0.0286±4e-4 | 0.696±6e-4 | 0.833±1e-3 | 0.0187±3e-5 |
| RNNf [47] | 0.269±6e-3 | 0.515±4e-3 | 0.0746±4e-4 | 0.517±9e-3 | 0.725±4e-3 | 0.0382±7e-4 | 0.472±9e-3 | 0.694±6e-3 | 0.0314±5e-4 | 0.733±2e-3 | 0.856±8e-4 | 0.0163±1e-4 |
| TERN [47] | 0.257±6e-3 | 0.510±4e-3 | 0.0729±6e-4 | 0.548±5e-3 | 0.746±6e-3 | 0.0358±5e-4 | 0.559±3e-3 | 0.752±2e-3 | 0.0265±2e-4 | 0.746±1e-3 | 0.865±4e-4 | 0.0154±8e-5 |
| NDT [78] | 0.287±3e-3 | 0.528±3e-3 | 0.0723±3e-4 | 0.565±2e-3 | 0.752±1e-3 | 0.0348±1e-4 | 0.575±4e-3 | 0.773±3e-3 | 0.0250±2e-4 | 0.756±3e-3 | 0.873±1e-3 | 0.0149±2e-4 |
| GWNet [76] | 0.272±8e-3 | 0.524±1e-2 | 0.0721±8e-4 | 0.606±4e-3 | 0.779±3e-3 | 0.0309±4e-4 | 0.588±2e-3 | 0.769±2e-3 | 0.0242±4e-4 | 0.724±1e-3 | 0.851±2e-4 | 0.0168±9e-5 |
| GS4-S [66] | 0.181±7e-3 | 0.463±2e-3 | 0.0792±7e-4 | 0.553±3e-3 | 0.749±3e-3 | 0.0350±3e-4 | 0.600±7e-3 | 0.782±1e-3 | 0.0236±3e-4 | 0.740±5e-3 | 0.861±3e-3 | 0.0158±3e-4 |
| GS4-G [66] | 0.268±3e-3 | 0.530±3e-3 | 0.0730±1e-4 | 0.586±7e-3 | 0.769±4e-3 | 0.0322±6e-4 | 0.659±3e-3 | 0.812±3e-3 | 0.0194±1e-4 | 0.753±8e-4 | 0.869±6e-4 | 0.0149±5e-5 |
| DCRNN [40] | 0.288±3e-3 | 0.545±4e-3 | 0.0707±7e-4 | 0.606±2e-3 | 0.782±2e-3 | 0.0302±2e-4 | 0.635±4e-3 | 0.797±2e-3 | 0.0208±3e-4 | 0.756±2e-3 | 0.870±9e-4 | 0.0148±1e-4 |
| **AMAG** | **0.331±4e-3** | **0.575±8e-4** | **0.0694±4e-4** | **0.657±2e-3** | **0.811±2e-3** | **0.0266±2e-4** | **0.665±2e-3** | **0.817±1e-3** | **0.0192±3e-4** | **0.763±4e-3** | **0.874±2e-3** | **0.0144±2e-4** |

over four datasets, compared to GWNet which has variable accuracy on different datasets. Similar trends can be observed in multi-step forecasting results. As in Table 2, Graph-based methods, including AMAG, demonstrate better learning of future neural dynamics, highlighting the potential of GNNs in this task. GS4-G performs better than GS4-S on all four datasets, indicating that GRU based temporal encoding could be more suitable for neural signal forecasting. We find that AMAG is more optimal by almost 10% in $R^2$ on Monkey C and B when compared with non-GNN methods. Other GNN based methods, such as DCRNN, perform well, but AMAG exhibits additional improvement. For example, on Monkey C, AMAG achieves $R^2$ score of 0.657, whereas DCRNN achieves around 0.606. Another advantage of AMAG is that it dynamically learns the pruning of unnecessary edges in the adjacency matrix, while DCRNN and variants of GraphS4mer rely on a predetermined correlation matrix to define the connected edges which is typically required to be sparse for optimal performance and handcrafted sparsification is needed. Thus, we prune the DCRNN, and GraphS4mer variants with K-Nearest Neighbour (KNN) pruning as mentioned in GraphS4mer such that each neuron is connected to half of the other neurons. Threshold pruning can be an alternative method for DCRNN, but we find that it is not optimal as the $R^2$ on Monkey C is equal to 0.573 and 0.445 for similarity thresholds of 0.5 and 0.8, respectively. These are lower than $R^2$ of 0.606 obtained with the KNN pruning.

In Fig. 3, we visualize the forecasted trajectory of the strongest baselines: DCRNN, GWNET, GS4-G, and AMAG. AMAG exhibits a higher closeness to the ground-truth signal (highlighted by black circles). This is consistent with behavior decoding from forecasted signals. Specifically, a linear behavior decoder is trained on ground truth neural and behavior signals (Monkey C). Then, the same decoder is used to decode behavior from AMAG and other GNN based baselines forecasted signals. Specifically, with the forecasted signals from the strongest baseline, we get decoding performance in terms of $R^2$ as follows: 0.350 (GS4-G), 0.350 (DCRNN), 0.440 (GWNet), 0.555 (AMAG), compared to 0.656 when using ground truth neural signals, demonstrating that AMAG achieves the best behavior decoding performance (0.555) compared to other GNN based methods regardless of a gap compared to using ground truth signal. Overall, the results demonstrate the advantage of AMAG over other methods and highlight the significance of utilizing GNN based methods for forecasting neuronal signals.

Table 3: ***Computation complexity estimation*** *for multi-step and one-step forecasting task. Bold marks AMAG estimates and those methods whose estimates are better than AMAG.*

| | AMAG | LRNN | LFADS | STNDT | NDT | RNNf | TERN | DCRNN | GWNet | GS4-G |
|---|---|---|---|---|---|---|---|---|---|---|
| | | | | Mutli-step Forecasting | | | | | | |
| #P(M) | **0.27** | 4.59 | 5.53 | 2.38 | 1.06 | 3.55 | 9.93 | 0.60 | 1.12 | 1.52 |
| T(S) | **9.74** | **2.18** | **3.44** | **4.70** | **5.96** | **7.22** | **8.48** | 11.01 | 12.27 | 13.53 |
| M(GB) | **5.74** | **0.12** | **0.15** | **0.19** | **0.04** | **0.11** | **0.34** | **1.69** | **1.53** | **4.48** |
| | | | | One-step Forecasting | | | | | | |
| #P(M) | **0.22** | 1.25 | 5.84 | 8.31 | 3.70 | 3.55 | 9.93 | **0.15** | 1.12 | N/A |
| T(S) | **9.90** | **1.97** | **3.29** | **4.61** | **5.94** | **7.26** | **8.58** | 11.22 | 12.54 | N/A |
| M(GB) | **5.36** | **0.04** | **0.15** | **0.43** | **0.10** | **0.11** | **0.34** | **1.05** | **1.53** | N/A |

**Investigating Adjacency Matrix Learned by AMAG.** We hypothesize that the learned adjacency matrix in AMAG indicates interaction strength between channels and channels with high interaction strength are vital in forecasting. To verify the assumption, we group the channels into High (H), Mid-high (M-H), Mid-low (M-L), and Low (L) according to connection strength in $Aa$. We estimate their importance as the degree of performance drop when channels in each group are masked in AMAG. As the testing of the generality of the channel importance, we performed the same procedure in NDT. The forecasting performance ($R^2$) of NDT and AMAG with masked channels as input are shown in Fig. 4 (A). Generally, masking of channels consistently leads to performance drop, particularly with more significant degradation of performance in both AMAG and NDT when masking happens in the H group. This confirms our assumption that the weights learned in adjacency matrices provide informative insights into channel importance. Additionally, we investigate the relationship between learned weights and channel proximity. Figure 4 (B) illustrates the projection of the interaction strength of a target channel (T) onto other channels in the space organized by the $\mu$ECoG arrangement. We observe that target channels exhibit stronger interactions with surrounding channels, which aligns with the properties of $\mu$ECoG arrays. These experiments collectively demonstrate that the learned adjacency matrix values are meaningful in terms of identifying important channels, and the learned strength of connections aligns with the spatial proximity among ECoG arrays.

**Neuron Trajectory Alignment.** Apart from learning meaningful adjacency matrices, it is also advantageous to study neural trajectories AMAG has learned during the forecasting task. For that purpose, we project the original neural signals and their hidden embeddings from AMAG into the 2D space with PCA in Fig. 4(C). We find that, while neural trajectories, projected from the original neural signals, are tangled with random start and endpoints, with the projection from the hidden embeddings of AMAG, the neural trajectories are disentangled for the corresponding time-step and exhibit a circular structure similar to [53]. This illustration indicates that AMAG learns informative features that can be visualized as coherent and interpretable neural trajectories in 2D space.

**Computation Complexity.** We additionally compare in Table 3 the number of parameters #P(M), training time per epoch T(S), and maximum memory cost during training M(GB). We observe that GNN based methods use much fewer parameters than non-GNN methods, with AMAG consistently using a minimal or comparable (DCRNN for one-step forecasting) number of parameters when compared to graph model baselines (DCRNN, GWNet, GS4-G). This demonstrates that the source of the contribution to AMAG performance is its architecture rather than the number of parameters. The graph architecture of AMAG provides the topological constraint across channels. The absence of the topological constraint in non-GNN based methods could result in overfitting with limited training samples. Adding regularization terms, e.g., weight decay could help, but will also limit models' capacity. We demonstrate it in Fig. 3 (B). Adding weight decay to NDT improves performance (orange triangle vs purple circle), however, as the attention dimension increases, the effect of regularization diminishes, and for dim>1024, weight decay does not contribute to further improvement.

## 4.3 Ablation Study

Here we examine how each component in AMAG contributes to forecasting. Specifically, we consider the following ablated versions of AMAG: (1) removing self-connection in AMAG (amAG), (2) removing both the "Add" and "Modulator" module (–AG), (3) Only sample dependent "Adaptor" is kept in "Add" Module ($\tilde{a}$MAG), (4) Adjacency matrix is not learnable (AM-G), (5) The "Add" is ablated from the graph operation (-MAG), (6) "Modulator" is removed (A-AG), (7) The "Adaptor" in excluded from the "Add" (aMAG). The results are shown in Table.4.

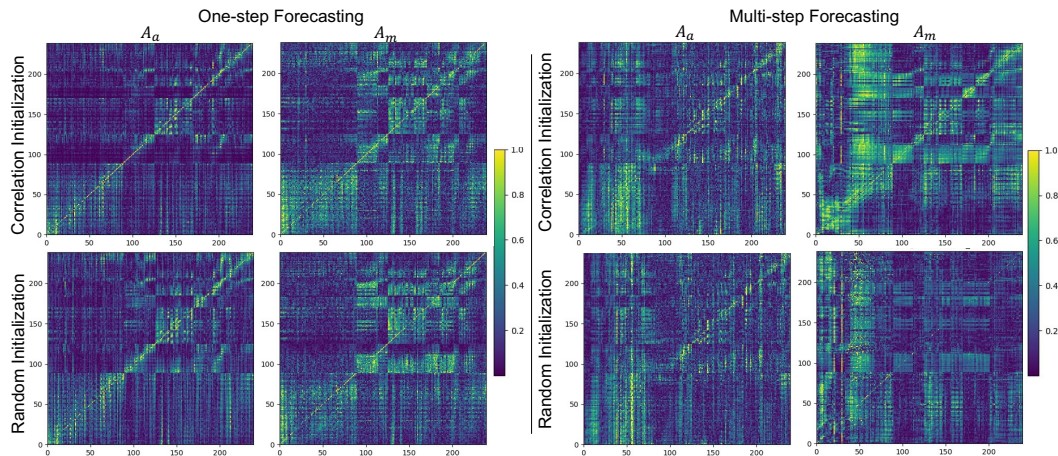

Figure 5: Visualization of the adjacency matrix of Add module ($A_a$) and Modulator module ($A_m$) learned by AMAG performing one-step forecasting (left) and multi-step forecasting (right) with correlation initialization (top) and random initialization (bottom).

We find that removing *Add* or *Modulator* modules can cause the most significant forecasting accuracy drop (–AG) as well as the self-connection part in the graph (amAG). When keeping sample-dependent Adaptor term only (ãMAG) in the Add module, the performance is very close to ablating the Add module (-MAG). The sample-dependent Adaptor term appears to be less important compared to other components but still reduces the accuracy if being ablated from AMAG (aMAG).

Table 4: *Comparison between the Ablated versions of AMAG.*

|  | $R^2 \uparrow$ | $CORR \uparrow$ | $MSE \downarrow$ |
|---|---|---|---|
| –AG | 0.424±1e-3 | 0.650±1e-3 | 0.0456±1e-4 |
| amAG | 0.427±6e-3 | 0.652±4e-3 | 0.0453±5e-4 |
| -MAG | 0.647±2e-3 | 0.805±6e-4 | 0.0274±2e-4 |
| ãMAG | 0.648±1e-3 | 0.806±4e-4 | 0.0273±6e-5 |
| A-AG | 0.652±9e-4 | 0.807±3e-4 | 0.0268±1e-4 |
| aMAG | 0.655±2e-3 | 0.810±1e-3 | 0.0269±2e-4 |
| AM-G Rand Init) | 0.575±2e-2 | 0.767±7e-3 | 0.0329±1e-3 |
| AM-G (Corr Init) | 0.617±2e-4 | 0.786±7e-4 | 0.0296±2e-5 |
| AMAG (Rand Init) | 0.652±1e-3 | 0.807±8e-4 | 0.0270±1e-4 |
| **AMAG (Corr Init)** | **0.657±2e-3** | **0.811±2e-3** | **0.0266±2e-4** |

In addition, we investigate the effect of initialization (random versus correlation) when the adjacency matrices are not adaptable (AM-G), specifically, AM-G (Rand Init 1), AM-G (Rand Init 2), AM-G (Corr Init). The forecasting accuracy of AM-G compared to the other two variants demonstrates the importance of correlation initialization. In particular, random and correlation initialization achieve similar performance when the adjacency matrices are adaptable, while random initialization leads to a less stable training process, see Appendix B.4. In Fig. 5, we visualized the learned adjacency matrix with correlation initialization (top) and random initialization (bottom) for one-step (left) and multi-step forecasting (right). Globally, the learned adjacency matrices show similar connection patterns, while locally, the connection can be different with different initialization matrices. This indicates that regardless of the initialization method, AMAG can converge to graph patterns with globally similar connections while multiple local connections are available to achieve similar forecasting performance.

## 5 Discussion

In conclusion, we propose AMAG for modeling spatial-temporal interaction between neural signals via forecasting future neuron activity. Experiments with both synthetic and neural signals show potential for discovering underlying neural signal interactions. The learned neural features by AMAG appear to disentangle in the PCA space. While adjacency matrix recovery is promising as training AMAG with one-step forecasting, more animal experiments could be done together with AMAG to understand the biological meaning of the learned adjacency matrix. It is worth noting that the memory requirement for GNN training increases as the number of neural recording channels increases. For future development, improvements to adapt the model for multi-region, multi-channel recordings could be considered.

# 6 Acknowledgements

Authors acknowledge the support in part by A3D3 National Science Foundation grant OAC-2117997 (ES,JL,TL,ALO), the Simons Foundation (LS,PR,ALO), the Washington Research Foundation Fund (ES), Departments of Electrical Computer Engineering (ES,JL,TL), Applied Mathematics (ES). Authors are thankful to the Center of Computational Neuroscience, and the eScience Center at the University of Washington. Authors would like to thank Rahul Biswas for discussions of synthetic dataset generation, Pan Li for discussions regarding GNNs, and the anonymous reviewers for insightful feedback.

# 7 Supplementary Information and Data

Supplementary information is available in the Appendix. Additional information and data will be made available through the project page `https://github.com/shlizee/AMAG`.

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

# Appendix A    Methods Details

In the following, we provide additional details regarding the initialization of the adjacency matrix, and temporal modeling (TE and TR components of AMAG).

Across the paper, we define the input to the model as $\boldsymbol{X} \in \mathbb{R}^{T \times C \times D}$, where $T$ denotes the total number of time steps, $C$ represents the number of channels in the neural recordings, and $D$ signifies the feature dimension. To refer specifically to the channel $v$ or the time step $t$ of input neural signals, we use the notations $\boldsymbol{X}^{(v)} \in \mathbb{R}^{T \times D}$ or $\boldsymbol{X}_t \in \mathbb{R}^{C \times D}$, respectively. To distinguish different data structures, matrices are denoted by bold capital letters, such as $\boldsymbol{X}$, $\boldsymbol{X}^{(v)}$, $\boldsymbol{X}_t$, while unbolded capital letters are used for vectors and scalars (e.g., $X_t^{(v)} \in \mathbb{R}^D$). $FC$ dentoes a single-layer fully connected neural network, $MLP$ denotes a multi-layer fully connected neural network, and $LN$ denotes the layer normalization technique. $\boldsymbol{A}_a$ and $\boldsymbol{A}_m$ are the adjacency matrices in Add and Modulator modules.

## A.1    Initialization of Adjacency Matrix

The adjacency matrices in AMAG are learnable parameters. We find that learning the adjacency matrix from scratch can be unstable. Hence we propose to initialize the adjacency matrix based on the correlation of each channel to stabilize the training. We flatten the features of channel $v$ and $u$, such that $X^{(u)}$, $X^{(v)} \in \mathbb{R}^{TD}$. Then the correlation between channel $u$ and channel $v$ is

$$Corr(u, v) = \frac{X^{(u)} X^{(v)T}}{\sqrt{|X^{(u)}||X^{(v)}|}}.$$

We then initialize each element in the adjacency matrices $A_a^{(u,v)}$ and $A_m^{(u,v)}$ as $Corr(u, v)$.

## A.2    Recursive Time Series Forecasting with TE and TR

Since Recurrent Neural Networks (RNN) are common in modeling continuous time-series signals, we employ a variant of RNN, Gated Recurrent Unit (GRU) [13], to facilitate the temporal encoding (TE) and readout (TR) modules. The TE module consists of a single-layer GRU that takes input from channel $v$ and computes the corresponding temporal embedding features $\boldsymbol{h}_t^{(v)} \in \mathbb{R}^d$ for each step $t$. These embedding features capture the underlying temporal dynamics of neural activity. To enable the exchange of information between the target channel $v$ and its neighboring channels $\boldsymbol{N}(v)$, specified by the adjacency matrices $\boldsymbol{A}_a$ and $\boldsymbol{A}_m$, the spatial interaction (SI) module updates the embedding features. This interaction among channels allows for modeling of spatial dependencies.

The TR module consists of a single-layer GRU and a fully connected ($FC$) layer. The module generates future neural signals. This pipeline of TE, SI and TR modules is shown below

$$\begin{aligned}
\boldsymbol{h}_t^{(v)} &= GRU_{TE}(X_t^{(v)}, \boldsymbol{h}_{t-1}^{(v)}), \ \ \boldsymbol{h}_0^{(v)} = \boldsymbol{0} \\
\boldsymbol{z}_t^{(v)} &= SI(\boldsymbol{h}_t^{(v)}, \{\boldsymbol{h}_t^{(u)} | u \in \boldsymbol{N}(v)\}, \boldsymbol{A}_a, \boldsymbol{A}_m) \\
\boldsymbol{r}_t^{(v)} &= GRU_{TR}(\boldsymbol{z}_t^{(v)}, \boldsymbol{r}_{t-1}^{(v)}) \\
X_{t+1}^{(v)} &= FC(\boldsymbol{r}_t^{(v)})
\end{aligned}$$

## A.3    Simultaneous Temporal Forecasting with TE and TR

The GRUs employed in TE and TR modules are particularly well-suited for recursive forecasting of future neural signals since they leverage the previously generated one-step signals to predict the next steps. In the multi-step forecasting scenario, however, where predictions for multiple future time steps are required, using GRUs in an autoregressive manner could lead to accumulation of errors. To address this issue and explore an alternative approach, we consider Transformers as an alternative to GRU. Transformers [68], simultaneously generate all future neural signals and could avoid the accumulation of errors. In this case, both TE and TR are implemented as decoder-based Transformers. The TR module predicts multiple future steps from a given initial time step $t$ up to the target time step $T$, based on the available neural recordings prior to $t$.

Before feeding neural signals into the Transformer model, several pre-processing steps are applied. These steps include a masking layer, a linear layer, and a positional embedding layer, which sequentially transform the neural signals into embedding space denoted as

$$\boldsymbol{E} = emb(\boldsymbol{X}) = FC(mask(\boldsymbol{X})) + PE.$$

The masking layer ($mask$) masks out all the future inputs, such that $\boldsymbol{X}_{t:T}$ is constant during multi-step forecasting. The linear layer ($FC$) transforms the input neural signals from their original dimension $D$ to an embedding space of dimension $d$ using a linear transformation ($FC : \mathbb{R}^D \Rightarrow \mathbb{R}^d$). Additionally, positional encoding ($PE$) is applied to incorporate positional information into the embedding space, The positional embedding is defined as $PE(pos, 2i) = sin(pos/10000^{2i/d})$, $PE(pos, 2i+1) = sin(pos/10000^{2i+1/d_{model}})$, as described in [68]. Here $pos$ corresponds to the position in the temporal space, and $i$ represents the position in the feature space. It is worth noting that channels of neural recordings are treated as feature dimensions when incorporating the positional embedding, ensuring that the Transformer can distinguish between different channels.

The embedded signal $\boldsymbol{E}$ is then mapped to $\boldsymbol{Q}_e^{(v)}$, $\boldsymbol{K}_e^{(v)}$, and $\boldsymbol{V}_e^{(v)}$ by weight matrix $\boldsymbol{W}_q, \boldsymbol{W}_k, \boldsymbol{W}_v \in \mathbb{R}^{d \times d}$ as Query, Key and Value in Transformer model. Specifically, $\boldsymbol{Q}_e^{(v)} = \boldsymbol{E}^{(v)} \boldsymbol{W}_q$, $\boldsymbol{K}_e^{(v)} = \boldsymbol{E}^{(v)} \boldsymbol{W}_k$, $\boldsymbol{V}_e^{(v)} = \boldsymbol{E}^{(v)} \boldsymbol{W}_v$. The outputs from the temporal encoding can be written as

$$\boldsymbol{ATT}^{(v)} = Softmax(\frac{\boldsymbol{Q}_e^{(v)}(\boldsymbol{K}_e^{(v)})^T}{\sqrt{d}})\boldsymbol{V}_e^{(v)},$$

$$\boldsymbol{H}^{(v)} = LN(MLP(\boldsymbol{E}^{(v)} + \boldsymbol{ATT}^{(v)})),$$

$$\boldsymbol{H}^{(v)} = [\boldsymbol{h}_1^{(v)}, \dots \boldsymbol{h}_T^{(v)}], \ \ \boldsymbol{H}^{(v)} \in \mathbb{R}^{T \times d}$$

Here we assume a single head in the Transformer, which could be extended to the multi-head situation as well. As in one-step forecasting, we update the feature with spatial interaction

$$\boldsymbol{z}_t^{(v)} = SI(\boldsymbol{h}_t^{(v)}, \{\boldsymbol{h}_t^{(u)} | u \in \boldsymbol{N}(v)\}, \boldsymbol{A}_a, \boldsymbol{A}_m),$$

$$\boldsymbol{Z}^{(v)} = [\boldsymbol{z}_1^{(v)}, \dots, \boldsymbol{z}_T^{(v)}].$$

TR module is similar to TE, with the difference that instead of receiving embedded signal as input, TR receives $\boldsymbol{Z}^{(v)}$ as input and generates $\hat{\boldsymbol{X}}^{(v)}$ with corresponding $\boldsymbol{Q}_z^{(v)} = \boldsymbol{Z}^{(v)} \boldsymbol{W}_q$, $\boldsymbol{K}_z^{(v)} = \boldsymbol{Z}^{(v)} \boldsymbol{W}_k$, $\boldsymbol{V}_z^{(v)} = \boldsymbol{Z}^{(v)} \boldsymbol{W}_v$. To clarify, we use the same notation for $\boldsymbol{W}_k$, $\boldsymbol{W}_q$, $\boldsymbol{W}_v$, but they are different sets of parameters in TE and TR. While we describe the process for generating $\boldsymbol{H}^{(v)}$ and $\boldsymbol{Z}^{(v)}$ for current channel $v$, it can be adapted to other channels. The parameters in TE and TR are shared across channels.

## Appendix B   Experiments

### B.1   Synthetic Dataset

**Experiments with the Synthetic Dataset.** In our experimentation with the synthetic dataset, we apply AMAG to learn four specific tasks:

- Reconstruction: In this task, the model aims to reconstruct neural signals without any masking or missing information. The entire input sequence is given as input to the model.

- Reconstruction with masking of the input: In this task, the model is trained to reconstruct neural signals while dealing with masked or missing information. The channels are masked in a blocked manner, where five consecutive channels at five consecutive time steps are randomly masked. The first $t$ steps are considered as the context and are not masked.

- Multi-step forecasting: In this task, the model is trained to predict multiple future steps from a given initial time step $t$ up to the target time step $T$.

- One-step forecasting: This task focuses on predicting the next step or a single future step based on the available input information.

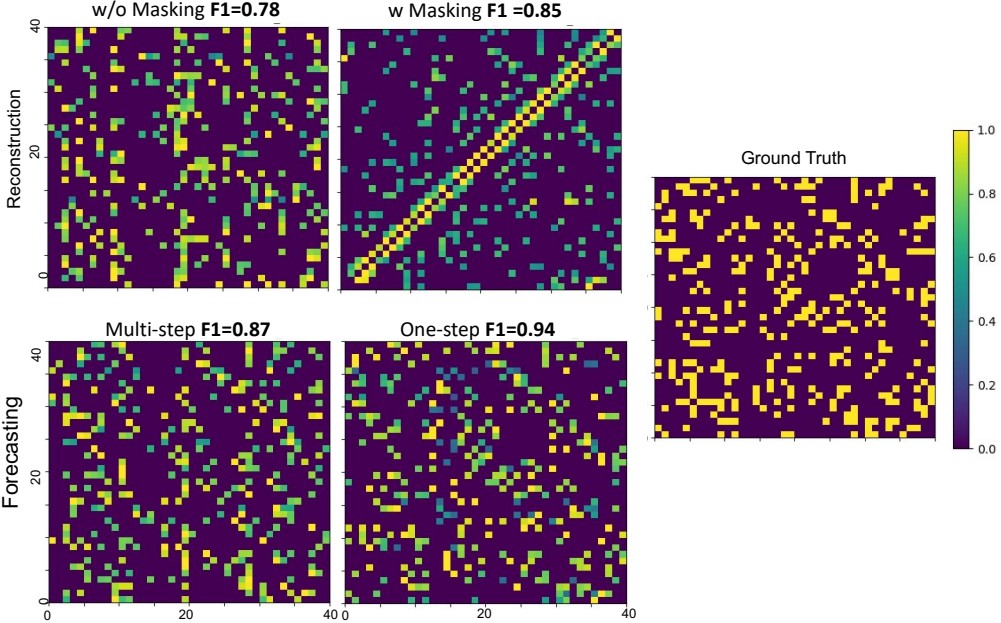

Figure 6: Adjacency matrices learned by AMAG visualized as colormaps and arranged by their F1 score (from left to right is from lower to higher), with the rightmost map indicating Ground Truth. Yellow color indicates full recovery of ground truth value (100% accuracy) while green-blue colors indicate recovery with lower accuracy. In particular, we show outcome adjacency matrices that AMAG recovers when solving left-right: the reconstruction task, the reconstruction of masked input task, multi-step forecasting task, one-step forecasting task, and ground truth.

To ensure a fair comparison between these tasks, we employ the same model structure, specifically using GRUs for both the TE and TR modules in all four cases, and compute the loss term for each task over the time steps from $t$ to the end of the sequence.

**Additional Results with the Synthetic Dataset.** In addition to the results in the main paper, we further assess AMAG ability to recover the learned ground truth adjacency matrix using the synthetic dataset, which consists of 40 channels with a connection ratio of 0.2. We visualize the learned adjacency matrices in Fig. 6, comparing the results obtained from AMAG computing various tasks: reconstruction, reconstruction with masking, multi-step forecasting, and one-step forecasting. The learned adjacency matrices have continuous values describing the strength of connections between channels. The color in the figure indicates the connection strength. We keep the top eight ($40 \times 0.2$) connections as the recovery of the ground truth connections for each channel. To evaluate the accuracy of the discovered adjacency matrices, we use the F1 score as a metric that measures the harmonic mean between precision and recall, providing a comprehensive assessment of the matrix recovery. F1 scores for the four considered tasks are ordered from low to high and are arranged from left to right in Fig. 6. A higher F1 score indicates a higher degree of accuracy in capturing the ground truth adjacency matrix, highlighting the effectiveness of each method in this discovery process.

## B.2 Neural Recordings Datasets

**Details of Monkey A and B Datasets.** As described in the main paper, we collected two new datasets from monkey Affogato (A), and Beignet (B) with $\mu$ECoG implanted in motor cortices, apply and compare AMAG and other methods with these recordings. All procedures were approved by the University of Washington Institutional Animal Care and Use Committee. The $\mu$ECoG covers five different subregions (Primary Motor Cortex (M1), Premotor Cortex (PM), Frontal Eye Field (FEF), Supplementary Motor Area (SMA), and Dorsolateral Prefrontal Cortex (DLPFC)) with 239 effective electrodes which have been previously used in [67]. During data collection, the subjects performed a reaching task towards one of eight different directions. The subject's hand position was tracked in

real-time using a marker-based camera system (Optitrak) and mapped to control the position of a cursor with which they performed the point-to-point movements. The collected neural signals can be segmented into trials based on the stage of the behavioral task. Within each trial, the subjects were trained first to reach the center target, followed by the appearance of the surrounding target. Subsequently, a delay period ranging from approximately 50ms to 150ms was introduced. After the delay period, the subjects moved from the center target to the surrounding target. A trial is considered successful if the subjects successfully reach the surrounding target, resulting in a reward. For our study, we include the neural signals recorded during the 600ms following the appearance of the surrounding target as our samples. Specifically, we treat the first 150ms of neural signals as a preparatory context for forecasting the future 450ms of neural signals. Consequently, trials with a duration shorter than 600ms after the appearance of the surrounding target are excluded from the study. In our study, we use data collected from monkeys A and B across 13 different sessions. These sessions are combined to form a comprehensive dataset for analysis. The dataset is split into 80% for training, 10% for validation, and 10% for testing. Specifically, dataset A contains 985 training samples, 122 validation, and 122 testing samples, while dataset B contains 700 training samples, 87 validation samples, and 87 testing samples. In experiments, we include all 239 electrodes from Monkey A and 87 electrodes, specifically from the M1 region of Monkey B. The neural time series for all train and test samples from monkeys A and B used in our analyses, along with meta-data of relative electrode locations within the array, will be shared to facilitate benchmarking of alternative algorithms.

**Details of Monkey M and C Datasets.** For dataset C and M [20, 64], we use neural signals recorded from the M1 of the two monkeys, including 96 channels. In these two datasets, we align the trials based on the movement onset, i.e., we include neural signals from 300ms before the movement onset and 450ms after the movement onset. Similarly, we split the datasets into 80% training data, 10% validation data, and 10% test data, collected in 6 sessions for Monkey C and 7 sessions for Monkey M. The number of training, validation and testing samples for Monkey C is 1285, 160, and 160, respectively. For dataset M, we have 1023 training samples, 127 validation, and 127 testing samples. All the trials have at least 450ms after the target onset.

**Details of Preprocessing.** M and C datasets contain the computed Local Motor Potential (LMP) and power band signal from eight frequency bands: 0.5–4 Hz, 4–8 Hz, 8–12 Hz, 12–25 Hz, 25–50 Hz, 50–100 Hz, 100–200 Hz, and 200–400 Hz. Full broadband data from A and B were collected at 25 kHz. We then applied a preprocessing procedure to extract LMP and powerband signals from A and B datasets as used in M and C datasets [20, 64]. All four datasets follow the normalization procedures to ensure consistency and comparability. Specifically, we normalize the data such that values for each channel within the range defined by mean minus four times the standard deviation ($mean - 4 \times std$) and mean plus four times the standard deviation ($mean + 4 \times std$) are linearly scaled to interval $[-1, 1]$. The mean ($mean$) and standard deviation ($std$) are computed for each channel based on the training set.

## B.3 Hyperparameters and Additional Model Details

To optimize the performance of models, we conducted a hyperparameter search by exploring different values for the learning rate, hidden size, and weight decay. Specifically, we considered learning rates of $10^{-3}$, $5 \times 10^{-4}$, and $10^{-4}$, hidden sizes of 64, 128, 512, 1024, and 2048, and weight decay values of 0, $10^{-4}$, $10^{-5}$, $10^{-6}$, and $10^{-7}$. We used the Adam optimizer to update the model parameters during training. We evaluate the model performance in terms of the $R^2$, correlation coefficient ($Corr$), and mean squared error ($MSE$) metrics on the validation set every 10 training epochs. The reported values in the tables of the main paper represent the testing set performance over the course of 500 epochs for most cases when the validation set achieves best performance. However, for the one-step forecasting task using the AMAG model and the STNDT-based models for both one-step and multi-step forecasting, we trained the models for 1000 epochs to ensure sufficient convergence and performance evaluation.

**Details of Models Used for Comparison**

- *LFADS* [65]. The LFADS model is a non-causal bi-directional GRU. We constrain the RNNs used in the LFADS to be uni-directional. Furthermore, since forecasting tasks typically require higher-dimensional features compared to reconstruction tasks, we set the factor dimension and the inferred input dimension to 256 for datasets A and B, and 64 for datasets M and C. The dimension

of the encoder, controller, and generator in the model is 512. The model is trained with an initialized learning rate of $10^{-4}$ reducing by 0.95 every 50 epochs and a weight decay rate of $10^{-4}$.

- *LRNN.* We build a one-layer RNN model. For one-step forecasting, we use the hidden dimension 1024 without weight decay on all four datasets. For multi-step forecasting, the hidden dimension is 2048 with a weight decay rate of $10^{-4}$ for M and C datasets and $10^{-5}$ for A and B datasets. All models are trained with the initial learning rate $10^{-4}$ decaying by 0.95 every 50 epochs.

- *RNNf* [47]. RNN_f is composed of a single-layer GRU. For one-step forecasting, the hidden dimension of GRU is 1024 with a weight decay rate $10^{-4}$ on M and C datasets and no weight decay for A and B datasets. For multi-step forecasting, we use hidden dimensions as 2048 on A and B datasets and 1024 on M and C datasets and set the weight decay rate to 0. The initial learning rate is $10^{-4}$, which decays by 0.95 every 50 epochs in all cases.

- *RNN PSID* [56]. We experiment with RNN PSID with the behavior-related hidden state of RNN PSID being 64 and the neuron-related hidden state being 512. The initial learning rate is $10^{-4}$. No weight decay is used in all cases.

- *NDT* [78]. The NDT model we use is a three-layer decoder-based Transformer. On A and B datasets, the attention dimension of the Transformer is 1024 for both multi-step and one-step forecasting. The attention dimension on M and C datasets is 512 and 256 for one and multi-step forecasting, respectively. Weight decay is set to 0 for one-step forecasting on four datasets. For multi-step forecasting, weight decay is $10^{-4}$ on the M and C datasets and $10^{-5}$ on A and B datasets. For all cases, we set the initial learning rate $10^{-4}$, which decays by 0.95 every 50 epochs in all cases.

- *STNDT* [37]. STNDT model we use has 10 layers of spatial-temporal blocks, each with one spatial attention Transformer and one temporal attention Transformer. The spatial Transformer has a hidden dimension as the temporal context length and the temporal attention has the dimension as the length of input neurons. We use the weight decay $10^{-4}$ for multi-step forecasting, $10^{-5}$ for one-step forecasting. We set the initial learning rate as $10^{-4}$ decaying by 0.95 every 50 epochs in all cases.

- *TERN*[1] [47]. We experiment TERN as a one-layer GRU followed by a Transformer layer, the hidden state of both GRU and Transformer is 1024, and the model is trained with an initial learning rate $10^{-4}$ decaying by 0.95 every 50 epochs. The weight decay rate is $10^{-6}$ for M and C datasets for both one-step and multi-step forecasting. Weight decay on A and B datasets is 0 and $10^{-4}$ and for one-step and multi-step forecasting.

- *GWNet* [76]. We use three layers GWNet where each layer has two blocks, including the three CNN kernels for dilation, residual, and skip convolution for A and B datasets. Before generating the future neural signals, there is one end CNN layer and one read-out CNN layer. The channel size is 64 for residual and dilatation CNN, 128 for skip CNN, and 64 for end CNN on A and B datasets. For M and C datasets, the channel size for residual, dilation, skip, and end CNNs are 128, 128, 256, and 512, respectively. To be noticed, for M and C datasets, each layer of GraphWaveNet has three blocks. The initial learning rate is $5 \times 10^{-4}$ decaying by 0.95 every 50 epochs. No weight decay is applied on A and B datasets, and $10^{-7}$ weight decay rate on M and C datasets. Kernel size is 2 for all CNN layers,

- *DCRNN* [40]. DCRNN we use is built as two diffusion convolutional layers, and each layer performs 2 diffusion steps. The hidden dimension of each diffusion layer is 64 for A and B datasets and 128 for M and C datasets, no weight decay is used. The learning rate is $5 \times 10^{-4}$ decaying by 0.95 every 50 epochs.

- *Graphs4mer* [66]. We use two variants of Graphs4mer, (i) with GRU (GS4-G) for temporal encoding, (ii)) with structured state space model (GS4-S), to perform the temporal encoding for all four datasets. The hidden dimension size is 256 on A and B datasets with a weight decay rate of $1e^{-5}$, and 128 on M and C datasets with a weight decay rate of 0. The learning rate is $5 \times 10^{-4}$, decaying by 0.95 every 50 epochs.

- *AMAG.* In a one-step forecasting scenario, we use single-layer GRU as a basic component for TR and TE with a 64-dimensional hidden size. For multi-step forecasting, we use Transformer layer as the basic component for TE and TR with hidden size 64. Sample-dependent matrix is computed using MLP taking the concatenated features from a pair of nodes as input, as used in

---

[1]TERN is also denoted as Neural RoBERTa, RoBERTa in short [32]

Table 5: ***Multi-step forecasting performance*** *of AMAG using GRU as the components for TE and TR modules (AMAG-G) versus AMAG using Transformer as the components for TE and TR modules (AMAG-T).*

|  | $R^2 \uparrow$ | $Corr \uparrow$ | $MSE \downarrow$ |
|---|---|---|---|
| **AMAG-G** | 0.653 | 0.807 | **0.0283** |
| **AMAG-T** | **0.658** | **0.811** | 0.0285 |

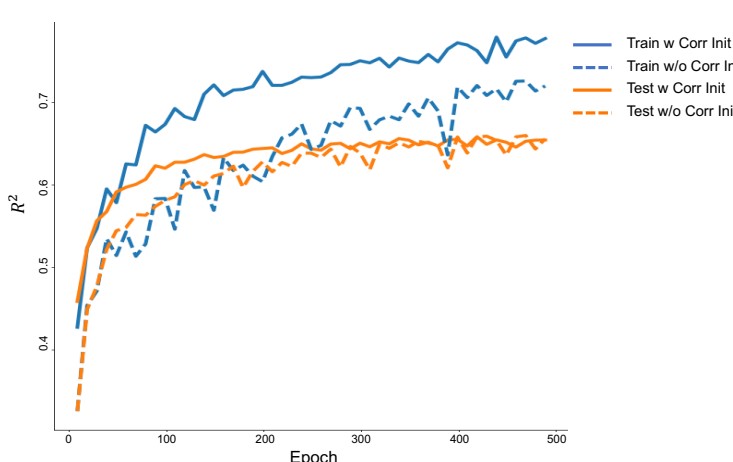

Figure 7: Visualization of $R^2$ score during the training and testing with the initialization of correlation matrix (solid curves) and random initialization (dashed curves). The correlation matrix initialized adjacency matrix stabilizes the training process.

[48]. The MLP encompasses four sequential fully connected layers whose input feature dimensions are $64 \times t$, $64 \times 2$, $64 \times 4$, and $64$ respectively where $t$ is the temporal context length. There is no weight decay on M and C datasets, $10^{-5}$ decay rate on A and B datasets. The initial learning rate is $5 \times 10^{-4}$ decaying by 0.95 every 50 epochs.

## B.4    Additional Results

**GRU and Transformer Comparison as Base Components for TE and TR for Multi-step Forecasting.** We compare how different temporal learning components for TE and TR change model performance in multi-step forecasting. We show the results in Table 5. We find that by incorporating the Transformer for temporal learning, AMAG could potentially achieve slightly better performance. We thereby use Transformer as the main components of TE and TR for multi-step forecasting.

**Correlation Matrix as the Initialization of the Adjacency Matrix in AMAG**. A well-initialized adjacency matrix in AMAG contributes to a more stable training process. We investigate two initialization approaches: random and correlation matrix initialization. Both initialization methods result in a similar performance in terms of the final $R^2$ scores, i.e., 0.659 (randomly initialized) *vs.* 0.658 (correlation matrix initialized). However, their learning curves exhibit distinct characteristics, as depicted in Fig. 7. When the adjacency matrix is initialized randomly (dashed curves), both training and testing curves display more fluctuations. In contrast, when the adjacency matrix is initialized with the correlation matrix (solid lines), the learning curves become smoother. This indicates that the correlation matrix initialization promotes a more stable and consistent learning process. Furthermore, examining the $R^2$ values in the initial epochs, we observe that the model initialized with the correlation matrix achieved higher performance compared to the random initialization. This finding suggests that careful initialization with the correlation matrix can accelerate the training process, leading to improved performance during the early stages of training.

**Adjacency Matrix Before and After Training.** We visualize the learned adjacency matrix alongside the correlation matrix in Fig. 8. We observe that the learned adjacency matrix is more sparse compared to the original correlation matrix. This indicates that during the training process, AMAG prunes and

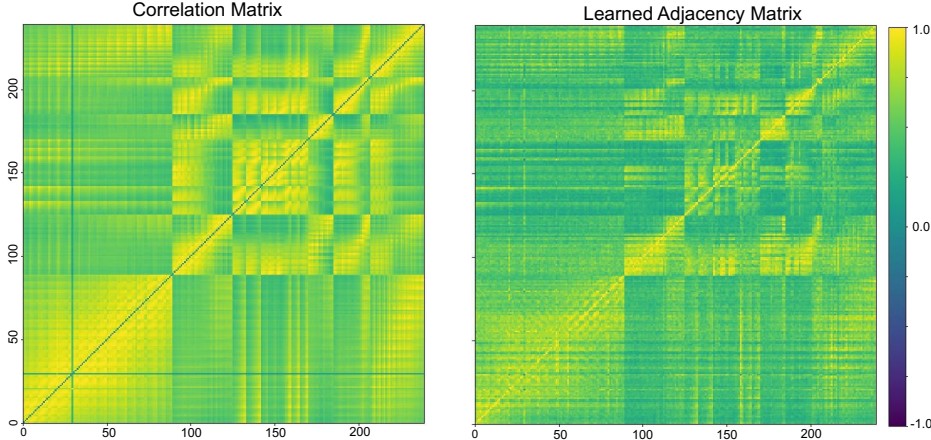

Figure 8: Illustration of correlation matrix as the initialization of the adjacency matrix in AMAG and the adjacency matrix after training. The learned adjacency matrix is more sparse compared with the correlation matrix (fewer values close to 1).

refines the correlation matrix, identifying the relevant and informative connections between neural channels for forecasting of neural signals.

## Appendix C   Analysis of Forecasting Error

We provide the following proofs to understand what would be the upper bound error of the predicted neural activity. Considering the complexity of the AMAG model study, the theoretical guarantee of the temporal forecasting tasks could be a potential research direction. As a starting point, we simplify the problem as follows: considering the one-step forecasting case and removing the nonlinearities in the model, the sample-dependent weight matrix, and the multiplicative weight matrix.

Assume we have neural signal recorded from $C$ channels at time step $t$, $\boldsymbol{X}_t \in \mathbb{R}^{C \times 1}$, output of TE layer, $\boldsymbol{H}_t \in \mathbb{R}^{(C \times D)}$. ($\boldsymbol{H}_0 = 0$). The $\boldsymbol{H}_t$ is updated with Linear RNN, such that $\boldsymbol{H}_t = \boldsymbol{X}_t W_1^e + \boldsymbol{H}_{t-1} W_2^e$, where $W_1^e$ and $W_2^e$ are weights of linear RNN. Outputs of SI and TR are represented as $\boldsymbol{Z}_t$ and $\boldsymbol{R}_t$ respectively.

$$\boldsymbol{Z}_t = \boldsymbol{A}\boldsymbol{H}_t$$
$$\boldsymbol{R}_t = \boldsymbol{Z}_t W_1^r + \boldsymbol{R}_{t-1} W_2^r$$
$$\boldsymbol{R}_0 = 0$$
$$\boldsymbol{Z}_t,\ \boldsymbol{R}_t \in \mathbb{R}^{(C \times D)}$$

We rewrite the $\boldsymbol{H}_t$ and $\boldsymbol{Z}_t$ such that they only depend on $\boldsymbol{X}$,

$$\boldsymbol{H_t} = \sum_{\tau=1}^{t} \boldsymbol{X}_\tau W_1^e (W_2^e)^{t-\tau}$$

$$\boldsymbol{Z_t} = \sum_{\tau=1}^{t} \boldsymbol{A}\boldsymbol{X}_\tau W_1^e (W_2^e)^{t-\tau}$$

$$\boldsymbol{R_t} = \sum_{\tau=1}^{t} \boldsymbol{Z}_\tau W_1^e (W_2^e)^{t-\tau}$$

$$= \sum_{\tau=1}^{t} \left( \sum_{q=1}^{\tau} \boldsymbol{A}\boldsymbol{X}_q W_1^e (W_2^e)^{\tau-q} \right) W_1^r (W_2^r)^{t-\tau}$$

The Predicted signal at the time $t+1$, $\hat{\boldsymbol{X}}_{t+1} = R_t \boldsymbol{W}$. Since we assume the model is stable, such that all the weight matrices should be bounded by M, i.e., $\|W_1^e\|_2, \|W_2^e\|_2, \|W_1^r\|_2, \|W_2^r\|_2 \leq 1$. Besides,

to make sure the RNN is stable, we need to constrain the eigenvalue of for $W_1^e, W_2^e, W_1^r, W_2^r$ to be smaller than 1, for example, $\forall \lambda_i \in \text{eig}(W_1^r)$, $\lambda_i < 1$. With the assumption, we remove the term $(W_2^e)^n$, $(W_2^r)^n$, $n > 2$ which corresponds to the contribution of previous t-2 steps to t+1 step prediction. Thus we have

$$\hat{X}_{t+1} = (AX_{t-1}W_1^e W_1^r W_2^r + AX_{t-1}W_1^e W_2^e W_1^r + AX_t W_1^e W_1^r)W.$$

We rewrite the $X_t$ and $X_{t+1}$ in form of early step plus a variation term, such that

$$X_t = X_{t-1} + \delta_1$$
$$X_{t+1} = X_t + \delta_2$$
$$= X_{t-1} + \delta_1 + \delta_2$$

The prediction error as $\epsilon = \hat{X}_{t+1} - X_{t+1}$,

$$\epsilon = (AX_{t-1}W_1^e W_1^r W_2^r + AX_{t-1}W_1^e W_2^e W_1^r + A(X_{t-1} + \delta_1)W_1^e W_1^r)W - X_{t-1} - \delta_1 - \delta_2.$$

Considering the first three terms that decide the $\epsilon$, we apply Cauchy–Schwarz inequality to get the upper bound,

$$\|(AX_{t-1}W_1^e W_1^r W_2^r + AX_{t-1}W_1^e W_2^e W_1^r +$$
$$A(X_{t-1} + \delta_1)W_1^e W_1^r)W\|_2 \leq 3\|A\|_2\|X_{t-1}\|_2 + \|A\|_2\|\delta_1\|_2$$

So we have:

$$\|\epsilon\|_2 \leq 3\|A\|_2\|X_{t-1}\|_2 + \|A\|_2\|\delta_1\|_2 + \|X_{t-1}\|_2 + \|\delta_1\|_2 + \|\delta_2\|_2$$

So if the signal is smooth enough that that $||\delta_1||_2 + ||\delta_2||_2$ is small, then the prediction error is bounded by the adjacency matrix $A$.

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
