# OpenReview forum: "AMAG: Additive, Multiplicative and Adaptive Graph Neural Network For Forecasting Neuron Activity"
_NeurIPS.cc/2023/Conference — NeurIPS 2023 poster_

### Official Review · Reviewer_zdkz · 2023-06-27

**Soundness:** 3 good
**Presentation:** 2 fair
**Contribution:** 3 good
**Rating:** 6
**Confidence:** 4

**Summary:**

This manuscript looks at time series forecasting in neural multi-channel electrical signals.  They propose a new graph neural network based approach called AMAG that infers connectivity between channels to improve the ability to forecast over competing approaches, most of which are designed to infer latent dynamics or predict relevant task related behavior.  There are multiple steps in this approach, and the entire system is trained to maximize multi-step prediction.  Experimental results suggest the ability to capture relevant connectivity.

**Strengths:**

This approach appears to accurately capture connectivity in synthetic signals, which appears to largely hold up in real world data as well.

Predictive performance shows large improvements in the forecasting (both one-step and multi-step) problem compared to competing approaches.

An ablation study implies that all components of the network are beneficial, motivating each component.

**Weaknesses:**

The major evaluation metric is forecasting in neural time series.  It is unclear how important this metric is by itself, whereas one often wants to evaluate how well we can extract relevant information (such as neural decoding) from the neural signals.  There is an implication that improving forecast of neural signals would improve these other problems, but it would be much stronger to explicitly show that.

My biggest hesitation is that it is unclear where the predictive improvements are coming from.  Most of the compared methods have performed well in many contexts, and the proposed structure is not so different than these existing methods.  The largest difference seems to be the GNN operating in the latent space of the system, whereas other methods capture these relationships in other ways, such as in the encoder function.  The ablation study does not clearly enough elucidate such differences, as all variants of the model seem to be extremely highly performing.  (I am assuming that Table 3 corresponds to Monkey C since the AMAG result matches up, but it doesn't explicitly state).  In fact, the ablated models still seem to be state-of-the-art, which is confusing.  I would encourage the authors to really explore the differences between their approach and existing methods and clearly show where the majority of the performance improvements happens.

There's a number of fuzzy details that need additional explanation throughout, such as the aforementioned ablation not clearly describing the data, and explicitly stating what the training of reconstruction and reconstruction with mask are in Figure 2.

**Questions:**

What component is responsible for the major performance improvements compared to existing approaches?

How does this approach do at capturing other information such as neural decoding?  Is there evidence that improving at this forecasting task will really improve other downstream tasks?

For AM-G with the non-learnable adjacency, do you use the initialized correlation?  How sensitive is performance to that choice?


**Limitations:**

The only thing that I would add is that the authors should discuss more clearly that their improvements and results are focused on forecasting in the discussion, and that results are not shown for many of the other tasks where competing models have been effective.  Other than that, the comments seem fair.

---

> ### Author Rebuttal · Authors · 2023-08-10
>
> **W1**. We thank the reviewer for pointing out the need for further information regarding the forecasting metric versus additional metrics. We focused on forecasting since the metric indicates the extent to which future signals, assumed unknown, could be predicted. For perfect forecasting accuracy, the unknown signals would be fully recovered. Note that in one-step forecasting, the accuracy in terms of R2 is close to such accuracy, while for multi-step, there is a significant gap still to overcome. There are also direct applications of estimating future activity, such as anomaly detection of neurons’ recording, reducing the latency of the BCI system, especially when the behavior signal itself is insufficient to predict future behavior, or there is no observable behavior. Examples include sleep spindles, epilepsy detection, and emotion detection, to name a few. We also investigated additional metrics, such as the reconstruction of connectivity.
>
> We agree that behavior decoding could be another useful metric. We examined how the forecasted signal (multi-step) can be used for behavior decoding. We trained a behavior decoder using linear regression on ground truth neural and behavior signals applied to Monkey C. Then we applied the decoder to AMAG and other baselines' forecasted signals to examine how well their signals can be used for behavior decoding. Our results show improvement in decoding compared to the baselines (below). We will add these results and discussion to the revised version of the manuscript.
>
> |    | GT    | GraphS4mer | DCRNN | GWNet | AMAG  |
> |----|-------|------------|-------|-------|-------|
> | R2 | 0.656 | 0.350      | 0.44  | 0.453 | 0.555 |
>
> **W2**.  As suggested by the reviewer, we included new variants in the ablation study by removing Self-connection (amAG) and removing both Add + Modulator (--AG) modules, i.e., the graph within AMAG, to elucidate the major component leading to SOTA forecasting (table below). The performance significantly drops for both amAG and –AG variants compared to full AMAG (R2=0.658).
>
> Combining the results with Table 3 in the main paper, we find that the two components: both the Self-connection module (amAG) and Add/Modulator (--AG) are essential for AMAG performance. Variants with only one module of the graph, i.e., only Add or only Modulator,  along with Self-connection lead to reasonable performance (-MAG=0.611, A-AG=0.616 ). The combination of all of them together provides the best accuracy. We will add these results and discussion to the revised version of the manuscript.
>
> |      | R2    | CORR  | MSE    |
> |------|-------|-------|--------|
> | amAG | 0.425 | 0.652 | 0.0486 |
> | —AG  | 0.423 | 0.649 | 0.0488 |
>
> **W3**.  In the ablation study, all results were obtained for the multi-step forecasting task (Monkey C).
>
> The reconstruction in Fig. 2 refers to: given an input neural signal, the model is trained to learn to reconstruct the exact input.
>
> Reconstruction with masking indicates that the input into the model is masked (i.e., has missing data) such that the model learns to both reconstruct the input and fill in the missing information. Please see Appendix Section 2.1 for further details re. the experimental design.
>
> **Q1**. We thank the reviewer for W1 and this question to elucidate the components responsible for the major functionality of AMAG. As discussed in W2, both Self-connection and at least one module of the graph: Add or Modulator, are essential. The combination of all three provides the best performance as Add and Modulator modules supplement each other.
>
> **Q2**. Indeed, the forecasting task is a fundamental task and the recovery of unknown future signals can facilitate related applications and decoding, e.g., behavior decoding. As shown in reply to W1, AMAG forecasted signal facilitates a more accurate prediction of future behavior (R2=0.555) than other methods (R2~=0.45).
>
> **Q3**. We do use correlation initialization in AM-G, and it plays a role when the adjacency matrix is non-learnable. For example, when we substitute correlation initialization with random initialization, R2 drops from 0.623 to 0.585  (see below). Furthermore, fixed random initialization is more sensitive to the choice than fixed correlation initialization as we show for two random initialization variants in the table below (Rand Init 1 and Rand Init 2) compared to correlation initialization (Corr Init).
>
> | AMAG versions      | R2    | CORR  | MSE    |
> |--------------------|-------|-------|--------|
> | AM-G (Rand Init 1) | 0.585 | 0.774 | 0.0346 |
> | AM-G (Rand Init 2) | 0.596 | 0.773 | 0.0336 |
> | AM-G (Corr Init)   | 0.623 | 0.811 | 0.0285 |

---

> > ### Comment · Reviewer_zdkz · 2023-08-15
> > **Thank you for your clarifications**
> >
> > These additional ablations are helpful. Thank you.

---

> > > ### Author Response · Authors · 2023-08-16
> > >
> > > We sincerely appreciate your positive feedback and your dedicated time to review our paper. We are pleased to know that we have successfully addressed your concerns.

---

### Official Review · Reviewer_BP1k · 2023-06-30

**Soundness:** 4 excellent
**Presentation:** 4 excellent
**Contribution:** 4 excellent
**Rating:** 7
**Confidence:** 2

**Summary:**

This paper presents AMAG — a graph neural network for modeling and forecasting neural population dynamics. The graph neural network has mechanism to describe the additive and multiplicative interactions between neurons, and a sample-dependent matrix to adjust the additive component. Experiments are carried out on both synthetic data and real neural signals, in comparison with several baselines.

**Strengths:**

This paper is well written and easy to follow. The rationale behind AMAG design is clearly described, especially the basis of utilizing prior knowledge about additive and multiplicative processes.

The results overall show an improvement of forecasting performance by AMAG, and the experimental results especially those on actual monkey data are very interesting.

The selected results on visualization of interaction strength and the neural population trajectory embedding, i.e, those in Fig 4, are also very informative and interesting.


**Weaknesses:**

1. The quantitative results listed (e.g., Table 2) lack sufficient statistics, especially considering that the margin of improvements in some metrics are quite small. Some measure of std would be necessary.

2. Other than the quantitative improvements (as shown in Table 1/Fig 3), the significance of such improvements can be better explained. E.g., in Fig 3, indeed it is evident that AMAG showed closer results to the true trajectory in the highlighted areas of the curve, however, in the remaining parts of the curve, it also falls short of capturing the true trajectories by a quite large margin (similar to the other methods), such as the big deflection in front of the highlighted area in the first curve of panel 2, or the deflection following the highlighted area in the second curve of the same panel -- Overall, it would seem that the improvements over other baselines are less significant or on par with the other major errors of the results. For signals that have a lot of temporal fluctuations, it would be good to understand what are the significance in minor to moderate level improvements in signal details when there are still substantial errors in the rest of the signals.

 3. While the key innovation of the paper is motivated by leveraging prior knowledge about the additive and multiplicative interaction among neurons, the results provided limited insights into whether this modeling hypothesis could be verified from the results. There are ablation studies to show that both components are important (which is a strength of the paper), but there lack additional analyses into what are the different interactions being learned (e.g., by looking into the two adjacency matrices).

4. Effect of the initialization on the two adjacency matrices should be elaborated with additional results (what happens when different correlation matrix is being used). What are the differences between the A_a and A_m after being initialized by the same matri?

**Questions:**

In addition to address the comments raised in the weakness section above, please also considering clarifying the following questions.

1. For the synthetic dataset presented in 4.1, what type of interaction mechanisms is used in the ground truth generation of data, i.e.., the function of G? Is it only additive, or multiplicative? The adjacency matrix A defines which interaction?

2. Similarly, in the investigations based on the adjacency matrix in 4.2, which A's are being examined (A_a for additive interaction, or A_m for multiplicative interaction)?

3. Overall, as mentioned in the weakness section, since the use of these two different interaction mechanisms is a key contribution in this paper, the authors should not only make it clear which A's are being considered (both in generation of data and in results), but it'd also be interesting if the authors add more analyses and results on how A_a and A_m may look like, and what are the underlying insights such differences or commonality could offer.


**Limitations:**

The authors discussed briefly the limitations associated with the study and future works necessary to address these limitations.

---

> ### Author Rebuttal · Authors · 2023-08-10
>
> **W1**. We thank the reviewer for recommending including statistics in Table 2.  We agree that adding multiple runs would elucidate the extent of variation and robustness of the results. Originally, we included runs as separate tables (one in the main paper and one in the Appendix). Following the reviewer’s recommendation, we included standard deviation of three runs for multi-step forecasting methods (As shown in Table R2 in the attached pdf). The improvement remains significant considering the standard deviation. We will update Table 2 and other tables with statistics in the revised paper.
>
> **W2**. The quantitative evaluation we examined, presented in Tables 1 and 3, is the collective evaluation comparing how predicted signals match the ground truth across multiple samples and multiple time points. Under the scenario, each sample and each time point equally contribute to the reported accuracy.
>
> The examples shown in the right panel of Fig 3 are forecasted signals in the multi-step forecasting scenario (Monkey C), where AMAG results in R2=0.658. LRNN and NDT, in this case, result in 0.507 and 0.567, respectively. Considering that all three methods do not match the ground truth perfectly, there could be deflection in each of them.
>
> However, when comparing within the highlighted time window of the first curve of panel 2, AMAG is closer to the ground truth (GT) than other methods. Whereas, at an earlier time to the highlighted time window, there is indeed a deflection.
>
> This deflection is for all three methods and close inspection shows that AMAG is slightly closer to GT than other methods. Similarly, for the second curve of panel two, AMAG is significantly better in the highlighted window, and in the window that follows it, all three methods appear to perform similarly. Overall, we observe that while there could be some windows in which all methods are similarly close or similarly deflect, there are windows in which AMAG has a significant advantage (these are the windows we highlighted). These improvements lead to an overall major improvement in the evaluation metric.
>
> **W3**. We thank the reviewer’s comment regarding further examination of the adjacency matrices for elucidation of primary components of AMAG. As suggested, we additionally visualized the learned adjacency matrices for both the Add module (Aa) and the Modulator module (Am) when AMAG is trained to perform one-step forecasting (Fig. R2 left panel) and multi-step forecasting (Fig. R2 right panel) on Monkey A. For each of the cases, we examined two scenarios to exclude initialization sensitivity: when both Aa and Am are initialized with the correlation matrix (Fig. R2 top) or both matrices are initialized randomly (Fig. R2 bottom). In the visualized matrix, the i-th row shows other channels’ contributions to the i-th channel.
>
> These figures show that each matrix by itself (Aa or Am) appears consistent across initialization and forecasting length. However, when comparinge Aa with Am, while there is resemblance in their global pattern (which supports that a single module could generate reasonable prediction), each has distinct local patterns. Am appears to be more dense, while Aa is more sparse. Looking at specific target channels, some channels appear to contribute more in the additive module, while other channels could be more important in the modulator module.
>
> **W4**. We thank the reviewer for suggesting studying the effect of different initialization. As described in W3, Aa and Am are different after the learning regardless of initialization methods. For example, when both two matrices are initialized with the correlation matrix for one-step forecasting ( top row of Fig. R2 left panel), Aa and Am show different patterns (as discussed in W3). The conclusion generalizes to the random initialized weight matrix and multi-step forecasting cases.
>
> Quantitatively, we experimented using random initialization of Aa and Am in Appendix (Fig.A2), showing that AMAG achieves similar performance using different initialization (0.659 with random init vs. 0.658 using correlation init), but the learning process of correlation initialized AMAG can be more stable (Fig. A2 in Appendix).
> With additional experiments on non-adaptable Aa and Am (AM-G) after initialization, correlation initialization can achieve better performance AM-G (Corr Init R2 = 0.623), compared to using random initialization AM-G (Rand Init 1 R2=0.585) (see below).
>
> | AMAG versions      | R2    | CORR  | MSE    |
> |--------------------|-------|-------|--------|
> | AM-G (Rand Init 1) | 0.585 | 0.774 | 0.0346 |
> | AM-G (Rand Init 2) | 0.596 | 0.773 | 0.0336 |
> | AM-G (Corr Init)   | 0.623 | 0.811 | 0.0285 |
>
> In summary, the adaptable Aa and Am will converge to different matrices and can achieve similar performance with different initialization. However, when Aa and Am are non-adaptable (AM-G), the performance of the model can be affected by initialization methods.
>
> **Q1**. To generate the synthetic data, we only use the additive adjacency matrix (Aa) as described in Eq. 2 of the main paper. In this case, adjacency matrix A defines the additive interaction.
>
> **Q2**. For the analysis Investigating the adjacency matrix learned by AMAG,  we examined Aa, i.e., additive interaction.
>
> **Q3**. As discussed in Q2, the analysis of the adjacency matrix in 4.2 is based on Aa for additive interaction. We also visualized Aa and Am learned for one-step and multi-step forecasting with both random initialized matrices and correlation-initialized matrices in the attached pdf (Fig R2) (as discussed in W3 and W4). We find that regardless of the initialization method, in the one-step forecasting scenario, Aa matrix converges to similar patterns. Similarly, Am also converges to similar patterns, however Aa and Am have similar global patterns and different local patterns. This indicates that different channels contribute differently in terms of additive and multiplicative interaction.

---

### Official Review · Reviewer_Byrz · 2023-07-02

**Soundness:** 3 good
**Presentation:** 3 good
**Contribution:** 3 good
**Rating:** 5
**Confidence:** 4

**Summary:**

This paper proposes a graph neural network with additive and multiplicative message passing operations for neuron activity forecasting. The proposed model AMAG consists of a temporal encoder (TE), a spatial interaction (SI) module, and a temporal readout (TR) module. TE and TR modules are sequence models such as GRU and Transformer. The SI module consists of Add and Modulator modules, which are motivated by the additive and multiplicative interactions between neurons. Experiments on synthetic data and real-world neural recordings suggest the superiority of AMAG compared to non-GNN and GNN baselines.

**Strengths:**

1. There is some originality in the design of the additive and multiplicative message-passing operations in the SI module.
2. The methods are technically sound.
3. Overall this manuscript is easy to understand.

**Weaknesses:**

1. A major weakness of the manuscript is in the experimental design: the authors only split the data into train and test sets, and the hyperparameters and best model were chosen based on the test set (Appendix 2.3). The model could have been overfitted on the test set and made the reported results questionable, even though the authors showed results on a different train-test split in the Appendix.
2. Comparisons to GNN baselines are not very fair. For instance, fully connected graphs are used for DCRNN and GraphS4mer. In the original papers for DCRNN (Li et al., 2017) and GraphS4mer (Tang et al., 2022), the graphs are pruned using a threshold and are therefore sparse. The authors also replace the S4 component in the baseline GraphS4mer with GRU, which is not the original GraphS4mer architecture.
3. Given that there is a major issue in the experimental design, I would not consider the contribution significant.

**Questions:**

1. Experimental design: Model selection should be done on a separate validation set, and the test set should be held-out for reporting results only. Please follow this best practice for experiments.
2. Baselines: Please have a more fair comparison for GNN baselines. e.g., use sparse graphs and use the original GraphS4mer architecture (S4 instead of GRU).
3. Why is $A_a$ needed in the Add module? Would sampled-dependent $S$ be sufficient? It would be interesting to see an ablation experiment where $A_a$ is not included.
4. Are the main results based on GRU or Transformer for TE/TR modules? Please clarify.
5. Why aren’t GNN baselines included in Table 1 for one-step forecasting?
6. Figure 4: Please explain how the trajectory in panel C is obtained.
7. Please show comparisons to the strongest baseline(s) in Figure 3 and Figure 4A.

**Limitations:**

Yes, limitations are discussed in Discussion. The authors do not foresee any negative societal impact.

---

> ### Author Rebuttal · Authors · 2023-08-10
>
> **W1** & **Q1**. We thank the reviewer for pointing out this concern for experiments set up. As the reviewer noted, we provided additional experiments in the Appendix on a second train-test split for all four datasets. The test set of the first experiment is effectively the validation set since we train the model with the same set of hyperparameters of the first train-test split without additional hyperparameter search on the second train-test experiment. Since we did not search for hyperparameters when the model was trained on the second training and testing set, the results shown in the Appendix should not be overfitted to the second test set.
>
> The results in the Appendix have the same trend as the results shown in the main paper, demonstrating that  AMAG successfully generalizes to a different set of training and testing data.
>
> Also, we performed hyperparameter search similar to AMAG  for other methods and used the same set of hyperparameters when training the model on the second train-test split. This is to ensure that the results are comparable.
> We will add an explanation of these details in the revised version of the paper.
>
> **W2** & **Q2**. We thank the reviewer for pointing out the need for further details regarding which graphs were used in baseline methods.  In both DCRNN and GraphS4mer we did use K-nearest neighbor (KNN) (as mentioned in GraphS4mer paper) pruned graphs.
> In GraphS4mer we pruned the graph such that each neuron is connected to half of other neurons. We also tried threshold pruning (also mentioned in GraphS4mer) but setting the threshold is sensitive, especially with a large threshold (in terms of similarity), feature smoothing loss could become “NaN.”
>
> For consistency, we pruned the graphs for DCRNN with KNN pruning. We additionally did experiments using threshold pruning with DCRNN on Monkey C with similarity thresholds of 0.5 and 0.8 where testing R2=0.573 and R2=0.445 (compared to DCRNN with KNN pruning being R2=0.618).
>
> We will add these details and further discussion of graph pruning in the revised version of the paper.
>
> We wish to point out that in regard to pruning, the advantage of AMAG is that AMAG automatically learns which connections should be kept instead of manually setting the sparseness of the model.
>
> The reason that we did not use S4 is because S4 is suitable for long sequences; Based on the experiment, the performance in terms of R2 becomes 0.556 (Graphs4mer-S4) when using S4 instead of 0.579 (Graphs4mer-GRU). In the final version of the paper we will also include variations of baselines and their results and add further discussion of each baseline.
>
> **Q3**. We thank the reviewer for the question. Aa is the primary weight matrix initialized with a correlation matrix that controls message passing between the channels without limitation in range. In contrast, each element in S is the output of the sigmoid function, ranging from 0 to 1 adapted for individual samples. S is viewed as the regularization term adjusting the weight in Aa matrix based on each input sample and is not sufficient by itself.
>
> When keeping S only we observed that the performance is very similar to removing both S and A - the included ablation results( approximately R2=0.605 on the Monkey C dataset). We will add this ablation variant to Table 3 in the revised version of the paper.
>
> **Q4**. We appreciate the reviewer's question regarding GNN baseline methods, such as DCRNN, GWNET, and GraphS4mer to one-step forecasting. These were originally designed for the multi-step forecasting scenario with the results in the original papers being for the multi-step forecasting.  Furthermore, since AMAG accuracy (R2, CORR) turned out to be close to 1, the one-step scenario would not be necessarily an ultimate test since other methods applied in this scenario could either be close to AMAG or lower than AMAG. We thus did not include the comparison of these methods with AMAG and focused on a more challenging (in terms of accuracy) scenario of multi-step forecasting. We will add these notes to the final version of the paper.
>
> As a result of the reviewer's question, we tried to adapt GNN baselines to one-step forecasting. Since GraphS4mer learns a dynamic graph for each time window (several steps), performing one-step prediction requires learning the dynamic graph for each step which could be redundant. DCRNN and GWNET could be adapted (results in the attached pdf Table R1). We find that all three compared methods obtain approximately similar results. AMAG is better on Monkey A, and B, and GWNET is better on Monkey M and C. In addition, to achieve similar performance, AMAG used 0.12 million parameters while GWNet used 2.2 million parameters. This discussion and results will be incorporated into the final version of the paper.
>
> **Q5**. The results in the main paper are based on GRU for one step and transformer for multi-step.
>
> **Q6**. First, we get each step feature using either original 96-channel neuronal recordings (96 dimensions) or hidden states (64*96-dimension outputs of SI). Then, we perform PCA on the concatenated features of all timesteps and all samples. Figure 4C shows the first 2 dimensions in the PC space for original neuronal recording (left) and hidden states (right).
>
> **Q7**. Replot Figure 3 in attached pdf Fig.R1 (A).
> The purpose of Figure 4A is to show that the SI module of AMAG is capable of selecting important channels. Specifically, testing if we mask channels with higher weight in the adjacency matrix (High importance) could cause a larger performance drop compared to masking channels with lower weight (Low importance). Thus comparisons are made per method. In addition, we examined the effect of masking the same set of channels of other methods, especially when the prior knowledge of channel interaction is not explicitly included in model design, i.e. NDT which was chosen since it has the same temporal structure as AMAG (using Transformer for temporal encoding).

---

> > ### Comment · Reviewer_Byrz · 2023-08-12
> >
> > Thank you for the replies and additional experiments.
> >
> > Re: response to Q1 & W1, I’m still not convinced that a second train-test split is sufficient. If there is an overlap in test set data between the first and the second train-test splits, the selected model hyperparameters based on the first test set could still work well for the second test set. And this is not the best practice in ML research.

---

> > > ### Author Response · Authors · 2023-08-16
> > >
> > > We thank the reviewer for the follow-up.
> > >
> > > Our intent in using two train-test sets, as described, was to make sure that, for variable size datasets and with a limited amount of samples, there would be enough samples for training. As we observed earlier, non-graph-based methods such as NDT can easily overfit to the training set, and while regularization such as weight decay could help, including more training samples would be more advantageous. Thus two train-test splits were chosen for evaluating these methods instead of the train-val-test split, and other models followed the same evaluation strategy.
> > >
> > > Considering the inherent nature of the random split, there indeed may be an overlap between the first and the second test sets (test-1 and test-2). The extent of the overlap accounts for approximately 10% of testing data across all four datasets. Since the overlap proportion is relatively small, the performance on test-2 primarily reflects models' performance on unseen data constituting 90% of unseen test-2 samples. Notably, this portion was not utilized for hyperparameter selection. Therefore, AMAG accuracy on test-2 is unlikely to be attributed to hyperparameters chosen based on the first test set.
> > >
> > > We further investigated this point by validating AMAG and GNN-based baselines with a train-val-test split. Namely, the validation set in this split is the same as test-1, i.e., val=test-1, such that the val set is the set on which hyperparameters were tuned. From the remaining samples, we randomly selected train and test sets, with the test set having the same number of samples as the val set.
> > >
> > > In Table D1, we report the performance on the new test set from this train-val-test split for multi-step forecasting. The table is to be compared with Table 2 (or Table R2, which includes variance) and Table A2. Compared across the three tables, the metrics do vary based on which test set was used for evaluation. This variation is per dataset, and its trend is mostly consistent with the order of the metrics that the methods achieve. E.g. for Monkey M, all metrics across methods worsen in Table D1 and Table A2 in comparison to Table 2 (R2). For Monkey A, all metrics across methods improve in Table D1 and Table A2 in comparison to Table 2(R2). \
> > > While there is variation between test sets in the values of the metrics, AMAG consistently achieves better accuracy than other methods regardless of which test set the methods are evaluated on. This holds on the train-val-test split as well.
> > >
> > >
> > > ---
> > > Table D1: Multi-step Forecasting on the New Test Set
> > > ---
> > >
> > >
> > >
> > > |            |    Monkey M    |                |                 |    Monkey C    |                |                 |    Monkey B    |                |                 |    Monkey A    |                |                 |
> > > |------------|:--------------:|:--------------:|:---------------:|:--------------:|:--------------:|:---------------:|:--------------:|:--------------:|:---------------:|:--------------:|:--------------:|:---------------:|
> > > |            | R2             | Corr           | MSE             | R2             | Corr           | MSE             | R2             | Corr           | MSE             | R2             | Corr           | MSE             |
> > > | GWNET      | 0.272$\pm$8e-3 | 0.524$\pm$1e-2 | 0.0721$\pm$8e-4 | 0.606$\pm$4e-3 | 0.779$\pm$3e-3 | 0.0309$\pm$4e-4 | 0.588$\pm$2e-3 | 0.769$\pm$2e-3 | 0.0242$\pm$4e-4 | 0.724$\pm$1e-3 | 0.851$\pm$2e-4 | 0.0168$\pm$9e-5 |
> > > | GraphS4mer | 0.267$\pm$3e-3 | 0.531$\pm$3e-3 | 0.0731$\pm$2e-4 | 0.586$\pm$7e-3 | 0.769$\pm$4e-3 | 0.0322$\pm$6e-4 | 0.659$\pm$3e-3 | 0.812$\pm$3e-3 | 0.0194$\pm$1e-4 | 0.753$\pm$8e-4 | 0.869$\pm$6e-4 | 0.0149$\pm$5e-5 |
> > > | DCRNN      | 0.288$\pm$3e-3 | 0.545$\pm$4e-3 | 0.0707$\pm$7e-4 | 0.606$\pm$2e-3 | 0.782$\pm$2e-3 | 0.0302$\pm$2e-4 | 0.635$\pm$4e-3 | 0.797$\pm$2e-3 | 0.0208$\pm$3e-4 | 0.756$\pm$2e-3 | 0.870$\pm$9e-4 | 0.0148$\pm$1e-4 |
> > > | AMAG       | **0.331$\pm$4e-3** | **0.575$\pm$8e-4** | **0.0694$\pm$4e-4**| **0.657$\pm$2e-38** | **0.811$\pm$2e-3** | **0.0266$\pm$2e-4** | **0.665$\pm$2e-3** | **0.817$\pm$1e-3** | **0.0192$\pm$3e-4** | **0.763$\pm$4e-3** | **0.874$\pm$2e-3** | **0.0144$\pm$2e-4** |

---

> > > > ### Comment · Reviewer_Byrz · 2023-08-19
> > > >
> > > > Thank you for the additional experiments. I would like to see the main results in the final version of the paper updated with proper train-validation-test splits. I have also revised my score.

---

> > > > > ### Author Response · Authors · 2023-08-21
> > > > >
> > > > > We will include the results of training-validation-test splits along with discussion regarding overfitting and regularization of non-GNN methods in the revised main manuscript. We appreciate the reviewer considering the clarifications regarding data splits and raising their score.

---

### Official Review · Reviewer_peT1 · 2023-07-07

**Soundness:** 3 good
**Presentation:** 3 good
**Contribution:** 2 fair
**Rating:** 6
**Confidence:** 2

**Summary:**

The authors introduce a graph neural network in their study to predict neural activity. This network comprises a temporal encoding and decoding layer specific to each channel, with a spatial interaction layer positioned in between. The inclusion of the explicit spatial interaction layer aids in capturing the underlying spatial relationships, as demonstrated using synthetic data. The proposed model exhibits state-of-the-art performance in both one-step and multi-step forecasting tasks when evaluated on actual neural data.
In terms of the model-readout, it incorporates both additive and multiplicative operations, with both being necessary to attain the demonstrated accuracy. This observation is supported by an ablation study.

**Strengths:**

- impressive enhancements in one-step forecasting performance
- notable improvements are observed in multi-step forecasting, although to a lesser extent
- explicit spatial interactions aid interpretability

**Weaknesses:**

- there is a presence of numerous non-standard acronyms in the paper that are not introduced upon their first mention
- the utilization of graph neural networks for analyzing neural signals is not entirely novel, this study showcases somewhat incremental progress in this field

**Questions:**

- What is PSID, what  is DCRNN? After the introduction section, subsequent acronyms are not explicitly introduced or defined.
- TE and TR can be transformer or GRU. Which one did you use for the results in the paper?
- Why is the improvement in multi-step forecasting comparatively moderate despite the impressive enhancements in one-step forecasting? Intuitively, one would expect the performance gap between methods to increase with an increasing number of forecasting steps.
- How many steps are used in multi-step forecasting?
- RoBERTa in Table 2 is not mentioned in the caption

**Limitations:**

The authors have discussed limitations of their work.

---

> ### Author Rebuttal · Authors · 2023-08-10
>
> **W1**. We appreciate the reviewer pointing out the use of non-standard acronyms in the paper. These are names of related methods introduced by their authors. We will make sure to include the full name of these methods at their first mention along with explanations of their origin, and provide citations in the revised version of our paper.
>
> **W2**. We agree with the reviewer that there have been previous works using graph neural networks for neural signals. But most of the previous works focus on neural signals recorded from EEG and fMRI where the temporal dynamics could be different from the field potential (Local Field Potential and ECoG) datasets that we study in this work. In addition, most of the previous works address the classification task, e.g., emotion detection or disease detection. Whereas we focus on the forecasting task.
>
> Forecasting of neural signals is important for both scientific understanding and application. Indeed, as demonstrated in Section 4.1 of the main paper, through learning to forecast, AMAG learns to recover the underlying interaction between channels. Furthermore, learning to forecast can be used for anomaly detection monitoring of the neuronal recordings and for reducing the latency of the Brain-Computer Interfaces (BCI) especially when future behavior is not simply related to past behavior. We show in this work that AMAG can achieve SOTA forecasting performance with a novel design of GNN architecture with both additive and multiplicative messaging passing operations.
>
> In addition to forecasting, we demonstrate that AMAG facilitates estimation of connectivity (on the synthetic dataset) and learned adjacency matrix on the experimental recordings dataset which shows meaningful ECoG arrangement (Figure 4 in main paper).
>
> **Q1**. We thank the reviewer for pointing out the need to define the acronyms early in the paper.
>
> PSID is the abbreviation of Preferential Subspace IDentification algorithm [54, 55], where the method identifies the behavioral relevant and irrelevant neuron subspaces by learning to predict the next step of neural signals and behavior signals in three stages. RNN PSID extends the algorithm with the RNN structure [55].
> DCRNN refers to Diffusion Convolutional Recurrent Neural Network [39] . The model combines the Diffusion Graph with GRU to perform traffic forecasting tasks. These methods were introduced and discussed in the Related Work section, which follows the Introduction.  In the revised paper, we will make sure we define them earlier to avoid confusion.
>
> References from the main paper:
> [39] Yaguang Li, Rose Yu, Cyrus Shahabi, and Yan Liu. “Diffusion convolutional recurrent neural 458 network: Data-driven traffic forecasting”. In: arXiv preprint arXiv:1707.01926 (2017).
> [54] Omid G Sani, Hamidreza Abbaspourazad, Yan T Wong, Bijan Pesaran, and Maryam M 503 Shanechi. “Modeling behaviorally relevant neural dynamics enabled by preferential subspace 504 identification”. In: Nature Neuroscience 24.1 (2021), pp. 140–149.
> [55] Omid G Sani, Bijan Pesaran, and Maryam M Shanechi. “Where is all the nonlinearity: flexible 506 nonlinear modeling of behaviorally relevant neural dynamics using recurrent neural networks”. 507 In: bioRxiv (2021).
>
> **Q2**. For one-step prediction, the results reported in the main paper are when both TE and TR are GRUs. For multi-step prediction, TE and TR are Transformers. The results for multi-step with GRU is R2=0.658, as shown in Appendix Table A3. We additionally ran one-step forecasting using Transformer and obtained R2=0.969. These results led us to choose TE, TR differently in one-step and multi-step forecasting.
>
> **Q3**. We would like to clarify that the improvement of AMAG vs non-GNN methods on one-step forecasting range from 0.06 to 0.09 in terms of R2 and comparable to other GNN methods. The improvement for the multi-step forecasting is dataset dependent, ranging from 0.01 to 0.09 in terms of R2 (comparing the same set of methods as in one-step forecasting Table 1).
>
> In most cases, the improvement is comparable between multi-step and one-step forecasting, except on Monkey A dataset, in which improvement of AMAG over other non-graph methods is more limited for multi-step forecasting.
>
>
> Such a situation is possible since better one-step forecasting does not guarantee better multi-step forecasting. One can think of the multi-step forecasting as a greedy search where even if the optimal solution at each step is obtained, it does not guarantee a global optimal solution in the multi-step case, but in many situations it will reach close to it.
>
> **Q4**. For all four datasets, in multi-step forecasting we predict future 15 steps, the period when monkeys are moving the cursor from the center target to the surrounding target.
>
> **Q5**. We thank the reviewer for noticing this typo. The label should be TERN (RoBERTa is the name that was used in an earlier work [31], while in a later work a similar model was renamed to TERN [46].) The label will be corrected in the revised version of the paper.
>
> [31 Bryan Jimenez. “Neural Network Models For Neurophysiology Data”. PhD thesis. Purdue University Graduate School, 2022
> [46] Ganga Meghanath, Bryan Jimenez, and Joseph G Makin. “Inferring Population Dynamics in Macaque Cortex”. In: arXiv preprint arXiv:2304.06040(2023).

---

> > ### Comment · Reviewer_peT1 · 2023-08-17
> >
> > Thank you for providing clarifications. I will await the discussion with the other reviewers before deciding whether to modify or maintain my score.

---

> > > ### Author Response · Authors · 2023-08-21
> > >
> > > We appreciate the reviewer’s feedback and will include the clarifications that the reviewer pointed out in the revised version of the manuscript.

---

### Official Review · Reviewer_Yw6R · 2023-07-12

**Soundness:** 3 good
**Presentation:** 3 good
**Contribution:** 3 good
**Rating:** 6
**Confidence:** 3

**Summary:**

The paper proposes a graph neural network-based model to forecast neural activities, which advances the DNN technology for neural understanding. It also proposes a method to leverage the causality structure of the signal into the moded design that generalizes the neural reconstruction task.

**Strengths:**

The paper uses a graph neural network to do future prediction tasks, which is more complex than a reconstruction task.

The method has been verified in neural signals from monkeys and shown to outperform other state-of-the-art methods, including LFADS, and TERN.

**Weaknesses:**

Some analysis on the importance of each module, self-connection, add module, and modulator module need to be discussed. For example, why are they all necessary?

Furthermore, the trade-off study between the model complexity and other sota methods is worth investigation.

Also, what is the computation bottleneck of the method, and how well does it scale?

Is there any theoretical guarantee that the model can predict the neural activity if the trajectory satisfies some regularity?

**Questions:**

It would be nice if the authors provided some theoretical analysis of why GNN based model can outperform the non-GNN-based model for neural forecasting.

Also, architecture complexity analysis is an interesting discussion.

**Limitations:**

limitations have been addressed

---

> ### Author Rebuttal · Authors · 2023-08-02
>
> **W1**. We thank the reviewer for suggesting additional analysis of the roles of Self-connection, Add and Modulator modules. These three modules are motivated by typical components in modeling neural activity, i.e. current activity, external additive input and gain modulation and according to our experiments are necessary to achieve AMAG accuracy as shown in the ablation study in Table 3 (main paper; Monkey C multi-step forecasting task). In particular, when Add or Modulator modules are ablated, R2=0.611 and R2=0.616, respectively, compared to R2=0.658 for full AMAG.
>
> We also ablated both the Self-connection in AMAG (amAG), where, in this variant, R2 drops to 0.425, elucidating the importance of the Self-connection module. This analysis will be added to the final version of the paper.
>
> **W2**. We thank the reviewer for suggesting to estimate and compare AMAG complexity. We  estimated the number of parameters, training time, and maximum memory cost (Table R3 in the attached pdf). Graph-based methods use much fewer parameters than non-GNN methods, with AMAG consistently using minimal or comparable (DCRNN for one-step forecasting) number of parameters when compared to graph model baselines (DCRNN, GWNet, GraphS4mer). Less parameters comes at the expense of computation time and memory cost. We will include these estimates and the discussion in W3 below in the revised version of the paper.
>
> **W3**. As presented above in W2, AMAG generally requires fewer parameters and less training time (compared to graph-based methods). The major bottleneck of AMAG is the requirement for larger memory, thus, scaling AMAG could be limited by the availability of working memory.
>
> **W4**. We appreciate the reviewer raising the important point of analyzing how the proposed AMAG model predicts neural activity. Analysis of complex, non-linear models, such as GNNs / AMAG for time-series data is not well determined, though there are some initial interpretations and explorations made [1, 2].
>
> While thorough analysis requires novel analytical tools and is outside of the scope of our work, under some simplifications analytical interpretation could be provided by simplifying AMAG to be linear and including only additive message passing and assuming that future signals rely on signals from two prior steps. Also, we constrain L2 norm of all weight matrices to be bounded by 1 for stability of linear RNN. Then upper bound error for $t+1$ can be expressed as
>
> $\| \hat{\boldsymbol{X}}\_{t+1} - \hat{\boldsymbol{X}}\_{t+1} \|
> \leq 3\|A\|\_2  \| \boldsymbol{X}\_{t-1} \|\_2 + \| A \|\_2\| \delta\_1 \|\_2  + \| \boldsymbol{X}\_{t-1} \|\_2 + \| \delta\_t \|\_2  \| + \|\delta\_{t+1} \|\_2$
>
> Here, $A$ represents the adjacency matrix, $\delta_1$ and $\delta_2$ indicate variations in signals between consecutive time steps. Thus, if channel interactions remain small ($\| A \|_2$ value), and signals exhibit smooth transitions (small $\| \delta_t \|_2 $) at each step, then the linear model will likely produce smaller prediction errors.
>
> We will include such simplifying scenarios in the Appendix of the revised version.
>
> [1] Agarwal, Chirag, et al., 2022.
>
> [2] Li, Yiqiao, Jianlong Zhou et al., (2022).
>
> **W5** & **Q1**. We agree with the reviewer that such analysis could be instrumental to further interpret GNNs and AMAG. While rigorous analysis of such high-dimensional and nonlinear systems requires novel advanced analysis tools and is out of the scope of the current study, we identify empirically that GNNs advantage primarily lies in the ability to capture the topology of the activity.
> Indeed, ECoG recordings exhibit spatial-temporal interactions among recording channels which could be represented as a graph. GNNs, by design, explicitly model the electrode interactions, thus including the prior inherent topology of the dataset. This prior knowledge could constrain GNN to be closer to the optimum. Indeed, if we completely remove the graph structure in AMAG (amAG), the accuracy drops to R2= 0.423 (Monkey C; multi-step forecasting).
>
> In non-GNN-based methods, channel signals are simply concatenated without topology information. The topology needs to be learned and thus typically these methods require more training samples. In datasets that we consider, the number of samples ranges from 874 to 1605. For such amounts of data, non-GNN methods, e.g., NDT, can overfit to the training data. Adding regularization terms, e.g., weight decay, could help, but will also limit the model's capacity. We demonstrate this in Fig. R2 (B) of the attached pdf by visualizing how R2 could be affected by the attention dimension and the weight decay. Adding weight decay to NDT training can improve (orange triangle vs purple circle). However, as the attention dimension increases, the effect of regularization diminishes, and for dim>1024 weight decay does not contribute to improvement.
> In contrast, GNN-based methods can be viewed as a type of  ‘adaptive regularization’ which constrains the model learning trajectory. We will add this discussion to the revised version of the manuscript.
>
> **Q2**. We measured the complexity of the model in terms of the number of parameters, training efficiency, and memory cost. As discussed in reply to W2, graph-based models typically require fewer parameters but longer training time compared to non-graph-based methods.
> The major computational bottleneck for AMAG is the requirement for larger working memory. One way to reduce memory cost and computation time would be to compress the original graph into a smaller hidden graph in the embedding space, with each node in the embedding representing multiple nodes in the original graph which is plausible considering that ECoG array records shared source input. Pruning could also reduce the graph size and reduce the memory cost, however, these methods will require manual setting of the pruning threshold and other hyperparameters.

---

### Author Rebuttal · Authors · 2023-08-10

We thank all reviewers for their insightful feedback.
We tried to address all the questions for each reviewer in the rebuttal session below with references to weakness (**W**) and questions (**Q**).

---

> ### Author Response · Authors · 2023-08-11
>
> We thank the reviewers for their positive view of our work and valuable feedback. We addressed reviewers' comments and questions in individual replies to each reviewer and attached pdf file. Please let us know if there are additional items or further clarifications/discussions we could address. We will incorporate clarifications and additions, as we specified in our replies, in the camera ready version of our work.

---

### Decision · Program_Chairs · 2023-09-21

**Decision:**

Accept (poster)

**Comment:**

This paper introduces AMAG, a graph neural network for predicting neuron activity, incorporating additive and multiplicative message passing operations. The model comprises a temporal encoder (TE), spatial interaction (SI) module, and temporal readout (TR) module. TE and TR leverage GRU and Transformer sequence models, respectively. The SI module integrates Add and Modulator modules inspired by neuron interactions. Comparative experiments using synthetic and real neural recordings demonstrate AMAG's superiority over non-GNN and GNN baselines.

The paper is well-structured and comprehensible. The rationale behind AMAG's design, including the incorporation of additive and multiplicative processes based on prior knowledge, is clearly elucidated. The results exhibit consistent enhancements in forecasting performance achieved by AMAG, particularly notable in the experiments involving actual monkey data. Overall, the paper contributes novel insights and advancements to the field.